# Measurement characteristics and genome-wide correlates of lifetime brain atrophy estimated from a single MRI

Anna E. Fürtjes [1,2] ✉, Isabelle F. Foote [3], Charley Xia [1,2], Gail Davies [1,2], Joanna Moodie[1,2], Adele Taylor[1,2], David C. Liewald [1,2], Paul Redmond[1,2], Janie Corley [1,2], Andrew M. McIntosh [4], Heather C. Whalley [4], Susana Muñoz Maniega [4], Maria Valdés Hernández[4], Ellen Backhouse[4,5], Karen Ferguson[4,5], Mark E. Bastin[4], Joanna Wardlaw [4], Javier de la Fuente [6], Andrew D. Grotzinger [3], Michelle Luciano [1,2], W. David Hill [1,2], Ian J. Deary [1,2], Elliot M. Tucker-Drob [6] & Simon R. Cox [1,2]

As a cardinal marker of brain ageing, lifetime brain atrophy obtained from a cross-sectional magnetic resonance image promises to boost statistical power to uncover novel genetic mechanisms of neurodegeneration. By analysing five young and old adult cohorts, we perform the most definitive study on lifetime brain atrophy's measurement and correlates. It is simply calculated from the relationship between total brain volume and intracranial volume, using the difference, ratio, or regression-residual method. Lifetime brain atrophy is correlated with well-validated neuroradiological atrophy ratings ($r = 0.37$–$0.44$), cognitive decline ($r = 0.36$), frailty ($r = 0.24$), and longitudinally-measured atrophic changes ($r = 0.36$). Lifetime brain atrophy computed with the difference method yields phenotypic and genetic signal similar to baseline intracranial volume ($r_g = 0.75$), in contrast to the residual method, which also best captures brain shrinkage. Lifetime brain atrophy is highly heritable ($h^2_{SNP} = 41\%[95\%CI = 38$–$43\%]$), and the strongest genome-wide association ($N = 43,110$) implicates *WNT16*, a gene linked with neuro-degenerative diseases.

The loss of brain matter due to ageing-related neurodegeneration, subsequently referred to as brain atrophy, is an important feature of non-pathological ageing[1], late-life cognitive decline (e.g.)[2], and accelerated neurodegenerative diseases such as dementia[3]. It is one of the cardinal neuroradiological markers of brain ageing. Brain atrophy can be observed in structural magnetic resonance imaging (MRI) scans, and is characterised by the widening of sulci, loss of gyral volume and the concomitant enlargement of ventricles as the brain shrinks away

from the intracranial vault and cedes space to increasing cerebrospinal fluid volume. As a dynamic within-person phenomenon, brain atrophy may be most adequately inferred with repeated brain MRI scans, through which individual-level brain matter volume changes can be tracked over time[4–6], and phenotypic and genetic predictors can be identified. However, the high costs and inconvenience of repeated MRI scanning can be prohibitive for longitudinal MRI data collection at large scale and over a sufficiently long time. Whereas there are several

[1]School of Philosophy, Psychology & Language Sciences, The University of Edinburgh, Edinburgh, UK. [2]Lothian Birth Cohorts, Department of Psychology, The University of Edinburgh, Edinburgh, UK. [3]Institute for Behavioral Genetics, University of Colorado Boulder, Boulder, CO, USA. [4]Centre for Clinical Brain Sciences, The University of Edinburgh, Edinburgh, UK. [5]UK Dementia Research Institute, The University of Edinburgh, Edinburgh, UK. [6]Department of Psychology, The University of Texas at Austin, Austin, TX, USA. ✉e-mail: afurtjes@ed.ac.uk

ongoing initiatives that will result in large-scale collection of longitudinal scans with other data (including genetic information), a reliable method of estimating brain atrophy from a single-occasion MRI scan of a person's brain would substantially boost sample sizes and statistical power for scientific discovery in the shorter-term. Given the burden of collecting data over lifespan periods, adopting such a single-occasion approach to modelling neurodegeneration may represent, in some cases, the only feasible option.

This is particularly true in the context of genome-wide association studies (GWAS) – for which many thousands of participants are required[7] – a measure of total brain atrophy obtainable from a single cross-sectional MRI scan is an especially attractive candidate for boosting statistical power. It could help uncover novel genetic signals and molecular mechanisms of neurodegeneration. Current meticulously collected longitudinal brain GWAS efforts remain understandably small and lacking in statistical power[8] ($N < 16,000$), given the rarity of longitudinal brain MRI studies with genetic data. Here, we present a phenotypic and genetic analysis of multiple approaches to derive an estimate of lifetime brain atrophy (LBA) from a single MRI scan.

In neuroradiological settings, it is common practice for trained experts to manually rate brain atrophy from a single MRI scan using well-validated visual scales[9–11]. Computational approaches concur with visual rating scales, but offer greater statistical power and greater fidelity to index adulthood atrophic changes[12]. LBA can be computed in middle-aged and older adults from a single-occasion brain MRI scan using measures of total intracranial volume (ICV) and total brain volume (TBV). Such an approach to LBA compares how much brain tissue an individual currently has (TBV) as some function of prior (or pre-neurodegenerative) brain volume—inferred from ICV[13–16]. Supported by evidence that ICV remains broadly stable across the lifespan[17–19]—which is not true for ageing-related reductions in the volume of the brain—one can employ ICV as an 'archaeological index' of an individual's maximum prior brain size (e.g., Ikram et al.[20]). Using this as a baseline, we may estimate LBA by comparing TBV and ICV in middle-aged and older adults. This concept is distinct from motivations in previous studies that adjust for ICV to control for potentially confounding factors associated with ICV (e.g., Wang et al.[21]), and it applies exclusively to older age cohorts that demonstrate some degree of brain matter shrinkage. Considering TBV with reference to ICV, rather than TBV alone, focuses the analysis on neurodegenerative processes as it amplifies associations with ageing-related traits[22]. Our reasoning does not apply to young cohorts because we cannot reasonably expect brain atrophy in young participants where TBV should be equal to (or very similar to) ICV (e.g., Dhamala et al.[23]). Our approach challenges previous assumptions that TBV in adults and head circumference in children are good proxies of one another[24].

However, there are different methods for comparing ICV and TBV to estimate a person's LBA. Here, we test three commonly-used computational approaches to derive LBA from ICV and TBV: the difference, ratio, and regression-residual methods (*Box S1*). Other studies refer to the *brain parenchymal fraction* when using the ratio method[16]. All three methods produce estimates with starkly different statistical properties, which has substantial implications for the resulting atrophy estimate and its interpretation. All three computational approaches have previously been used (e.g., Brouwer et al.[8] used difference method; Wang et al.[21] used ratio and residual methods to adjust for ICV), but to our knowledge no study has provided a side-by-side evaluation of the resulting measurement characteristics. It is unclear which method most clearly reflects age-associated brain shrinkage or longitudinally-observed atrophic changes, and whether the methods differentially account for ageing-related health traits. Furthermore, it remains unknown whether the different approaches capture variance associated with distinct genetic signal, and distinct genetic overlap with other ageing-related traits. Specifically, there is a need to establish

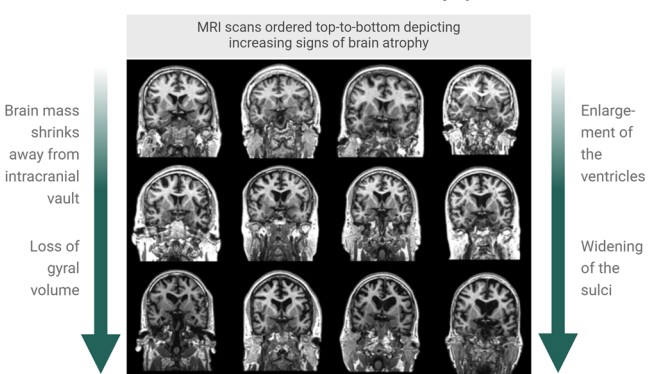

**A** Cross-sectional vs. longitudinal measures of brain atrophy

**Lifetime brain atrophy (LBA)**
- estimated from the divergence between the volume of the intracranial vault (*ICV*) and total brain volume (*TBV*)
- requires one cross-sectional MRI scan
- indexes lifetime atrophy, i.e., any brain volume ever lost, potentially over a decade-long time-span

**Longitudinal atrophic changes**
- observed as the divergence between total brain volume at MRI scan 1 (*TBV₁*) and total brain volume at MRI scan 2 (*TBV₂*)
- requires two repeated MRI scans (at time 1 and time 2)
- indexes atrophy that occurred between MRI scan 1 and 2

**Three computational methods**

| Difference score | ICV minus TBV |
| Ratio score | TBV divided by ICV |
| Residual score | TBV regressed onto ICV |

**Three computational methods**

| Difference score | TBV₁ minus TBV₂ |
| Ratio score | TBV₂ divided by TBV₁ |
| Residual score | TBV₂ regressed onto TBV₁ |

**B** **Markers of lifetime brain atrophy**

MRI scans ordered top-to-bottom depicting increasing signs of brain atrophy

Brain mass shrinks away from intracranial vault

Loss of gyral volume

Enlargement of the ventricles

Widening of the sulci

**Fig. 1 | Atrophy estimation from cross-sectional and longitudinal magnetic resonance imaging scans. A** Illustration of the differences between cross-sectionally-estimated lifetime brain atrophy (LBA) and longitudinally-observed atrophic changes and the three distinct computational methods used to calculate brain atrophy (top panel). Illustration was created in BioRender. Fürtjes, A. (2025) https://BioRender.com/wyju63y. **B** The bottom panel shows MRI scans ordered top-to-bottom depicting increasingly strong signs of brain atrophy. Both visual rating scales and computational methods can be applied to estimate brain atrophy from a single neuroimaging scan such as this. Coronal and axial brain MRIs are T1-W slices of 73 year-old adults from the Lothian Birth Cohort 1936, adapted from Cox, S. R., & Deary, I. J. (2022). Brain and cognitive ageing: The present, and some predictions (…about the future). Aging Brain, 2, https://doi.org/10.1016/j.nbas.2022.100032 with permission under CC-BY 4.0.

whether using the difference method mainly captures polygenic signal associated with baseline ICV, and whether the inclusion of ICV (or another strongly related proxy of skull size) as a covariate in a TBV GWAS captures genetic variance associated with brain atrophy rather than just initial brain or overall head/skull size.

In this pre-registered study (https://osf.io/gydmw/) we leveraged five large-scale cohorts to conduct the largest genetic study to-date of LBA in middle- and older adulthood (Fig. 1). First, we phenotypically validated LBA, showing it was substantially associated with neuroradiological atrophy ratings. Out of the three computational approaches, the residual method captured maximal associations with cognitive decline, frailty, age-associated brain shrinkage, and longitudinally-measured atrophic changes. Second, we performed the

largest LBA genome-wide association study (GWAS) to-date ($N = 43,110$) and show that LBA as a residual exhibited substantial genetic signal independent of ICV; it was highly heritable and had strong polygenic signal. The most significant GWAS genomic risk locus implicated *WNT16*, a gene previously linked with neurodegeneration. In contrast, LBA as a difference score mostly indexed ICV, suggesting that it has far-reaching implications for future longitudinal and cross-sectional GWAS studies of brain ageing that modelling change with a difference score approach mainly captured baseline variance (i.e., skull size) rather than brain matter shrinkage.

## Results

### Description and characterisation of the LBA phenotype

It is the overarching aim of this paper to validate measures of LBA estimated from a single MRI scan using the difference (*ICV-TBV*), ratio (*TBV/ICV*), and regression-residual method (TBV - ICV). Phenotypic analyses are structured according to the following hypotheses: Section (1) we reasoned that LBA should be associated with atrophy rated by neuroradiologists, and other brain- and ageing-related traits. Those analyses characterise the three LBA phenotypes, and the associations are calculated without additional covariates for a straightforward comparison. Section (2) we hypothesised LBA to increase with age, which we investigated with Pearson's correlations between LBA and age across multiple cohorts that were subsampled with increasingly strict age cut-offs. Section (3) we reasoned that LBA should overlap with atrophic changes observed in longitudinal data. Pearson's correlations between LBA extracted from cross-sectional data (at age 82 years in LBC1936; second neuroimaging visit in UKB) and atrophic changes observed from longitudinal data indicate the extent to which their captured variance overlaps. Longitudinally-observed atrophic changes were modelled using the same computational approaches as LBA (i.e., difference, ratio, residual method) to estimate discrepancies in TBV between the first and last available MRI scan; that is, between ages 71 and 82 in LBC1936, and between first and second neuroimaging visit in UKB. *Supplementary Data* depict descriptive statistics (Supplementary Data 6), variable distributions coloured by age across five cohorts (MRi-Share, Human Connectome Project [HCP], UK Biobank [UKB], Generation Scotland, and LBC1936; Fig. S3), as well as Pearson's correlations across TBV, ICV, cerebrospinal fluid (CSF) volume and LBA derived with the difference, ratio, and residual method (e.g., correlation between ICV and LBA in LBC1936: $r_{difference} = 0.81$ [95%CI = 0.76–0.84], $r_{ratio} = 0.49$ [95% CI = 0.40–0.57], $r_{residual} = 0.00$ [95%CI = −0.12–0.11]; Figs. S4, 5). Overview over performed analyses are in Supplementary Data 3. LBA $_{residual}$ and LBA $_{ratio}$ were flipped, hence, larger LBA values represent more brain atrophy. Analyses in LBC1936 confirm that LBA extracted from different FreeSurfer versions ($N = 581$) or through manual segmentation ($N = 631$) yielded substantially correlated variance, but that using FreeSurfer eTIV to approximate ICV made LBA $_{residual}$ biased towards larger manually-segmented ICV (*Methods 'MRI processing robustness checks in LBC1936'*).

### Measures of LBA predict brain atrophy rated by neuroradiological experts, as well as other ageing-related health traits such as frailty and cognitive ability

Since the difference, ratio, and residual method produced LBA scores with quite different statistical properties (*Box S1*), we compared their associations with visually rated atrophy and other ageing-relevant phenotypes in two cohorts with repeated MRI measures: the LBC1936 ($N = 286$, 71–73 years of age at baseline) and the UK Biobank (UKB; $N = 4,674$, 46–81 years of age). Figure 2 summarises associations in LBC1936 and Fig. S7 in UKB. Longitudinally-collected data assessing various cognitive tests, frailty, and body mass index (BMI) were modelled using growth curve models. Intercepts (e.g., *iCog*) were extracted to capture variance associated with stable baseline

characteristics at age 71, and slopes (e.g., *sCog*) were extracted to capture variance associated with change between the ages 71–82 years.

First, we externally validated LBA by demonstrating substantial associations between LBA and brain atrophy rated by well-validated[25] neuroradiological rating scales (*beta* $_{residual} = 0.46$ [95%CI = 0.32–0.60], *beta* $_{ratio} = 0.53$ [95%CI = 0.39–0.66], *beta* $_{difference} = 0.50$ [95% CI = 0.37–0.63]). Such visual rating scales were used by trained experts to manually rate brain atrophy from a single MRI scan in LBC1936 (description in *Supplementary Materials*). Associations between longitudinally-observed atrophic changes and visually rated atrophy were smaller (*beta* $_{residual} = 0.26$ [95%CI = 0.12–0.41], *beta* $_{ratio} = 0.26$ [95% CI = 0.11–0.41], *beta* $_{difference} = 0.25$ [95%CI = 0.10–0.39]), and there was no association between TBV and visually rated atrophy (*beta* $_{TBV} = -0.07$ [95%CI = −0.22–0.07]). Integrating TBV and ICV directly in the model [*lm(outcome ~ TBV + ICV)*] had a variance inflation factor below 5 (VIF =- 2.25) and marginally increased effect sizes compared with a two-step residual approach [LBA $_{residual} = resid(lm(TBV + ICV));$ *lm(outcome ~ LBA $_{residual}$)*] (SFig. 40). Second, associations with other ageing-related traits were similarly strong for LBA (Fig. 2A, C) as they were for longitudinally-observed atrophic changes (Fig. 2B, D) in LBC1936. For conciseness, below we only discuss associations for LBA $_{residual}$, which was similar to those from LBA $_{difference}$ and LBA $_{ratio}$. Greater LBA explained sizable percentages of variances in, and was associated with lower cognitive ability (i.e., *iCog* in Fig. 2; *beta* $_{residual} = -0.29$ [95%CI = −0.40 – −0.18]) as well as steeper rates of cognitive decline (i.e., *sCog*; *beta* $_{residual} = -0.36$ [95%CI = −0.47 – −0.25]), greater baseline frailty (i.e., *iFrailty beta* $_{residual} = 0.24$ [95% CI = 0.13–0.35]), greater longitudinal increases in frailty (i.e., *sFrailty beta* $_{residual} = 0.21$ [95%CI = 0.09–0.32]), and a larger brain age gap (i.e., brains appearing older given chronological age; *beta* $_{residual} = 0.42$ [95% CI = 0.31–0.53]). Given participants in the LBC1936 were assessed at the age of 73 years, the cognitive intercept variable (*iCog*) likely captured not only baseline levels of cognitive function, but also decades-worth of cognitive decline. It validates LBA $_{residual}$ as a marker of cognitive and brain *ageing* that it predicted the cognitive intercept significantly (*beta* $_{residual} = -0.29$ [95%CI = −0.40 – −0.18]) when LBA $_{difference}$ (*beta* $_{difference} = -0.02$ [95%CI = −0.13–0.10]; *ns.*) and LBA $_{ratio}$ did not (*beta* $_{ratio} = -0.16$ [95%CI = −0.28 – −0.05]; *ns.*). There was a significant aggregated trend to support that LBA $_{residual}$ explained more variance ($R^2$) across all estimates than LBA $_{difference}$ ($p = 0.0001$; not LBA $_{ratio}$ $p = 0.085$). Though, this was only the case when excluding outlier traits (Cook's distance > 4/15), which were visually rated atrophy (deep and superficial) and brain age; the three cross-sectionally-derived neuroimaging traits included in this analysis (Fig. S11; Fig. S13 for UKB). Associations were largely consistent when analysing males and females separately, but the association between LBA and *sFrailty* was driven by females (Fig. S9).

In the same LBC1936 participants, those associations differed when the predictor variable (i.e., LBA) was replaced with longitudinal atrophic changes (observed between ages 73–82 years) in that observed atrophic changes only indexed inter-individual *changes* across this same 9 year period (i.e., slopes rather than intercepts): Those 9 year atrophic changes were associated with steeper rates of cognitive decline (*beta* $_{residual} = -0.35$ [95%CI = −0.44 – −0.27]) and steeper rates of worsened frailty (*beta* $_{residual} = 0.25$ [95% CI = 0.14–0.37]), but not baseline levels of cognitive ability or frailty (i.e., intercepts). Clinically-ascertained all-cause dementia[26] was less strongly associated with LBA ($p = 0.011$ did not survive correction for multiple testing; *OR* $_{residual} = 1.72$ [95%CI = 1.13–2.61]) than it was associated with the longitudinal 9 year atrophic changes (*OR* $_{residual} = 3.34$ [95%CI = 1.98–5.65]). This may reflect that dementia patients exhibit the steepest rates of disease-caused brain shrinkage in years proximal to diagnosis– which we capture more directly with longitudinal MRI measures across the 9 year period. By contrast, the broader LBA

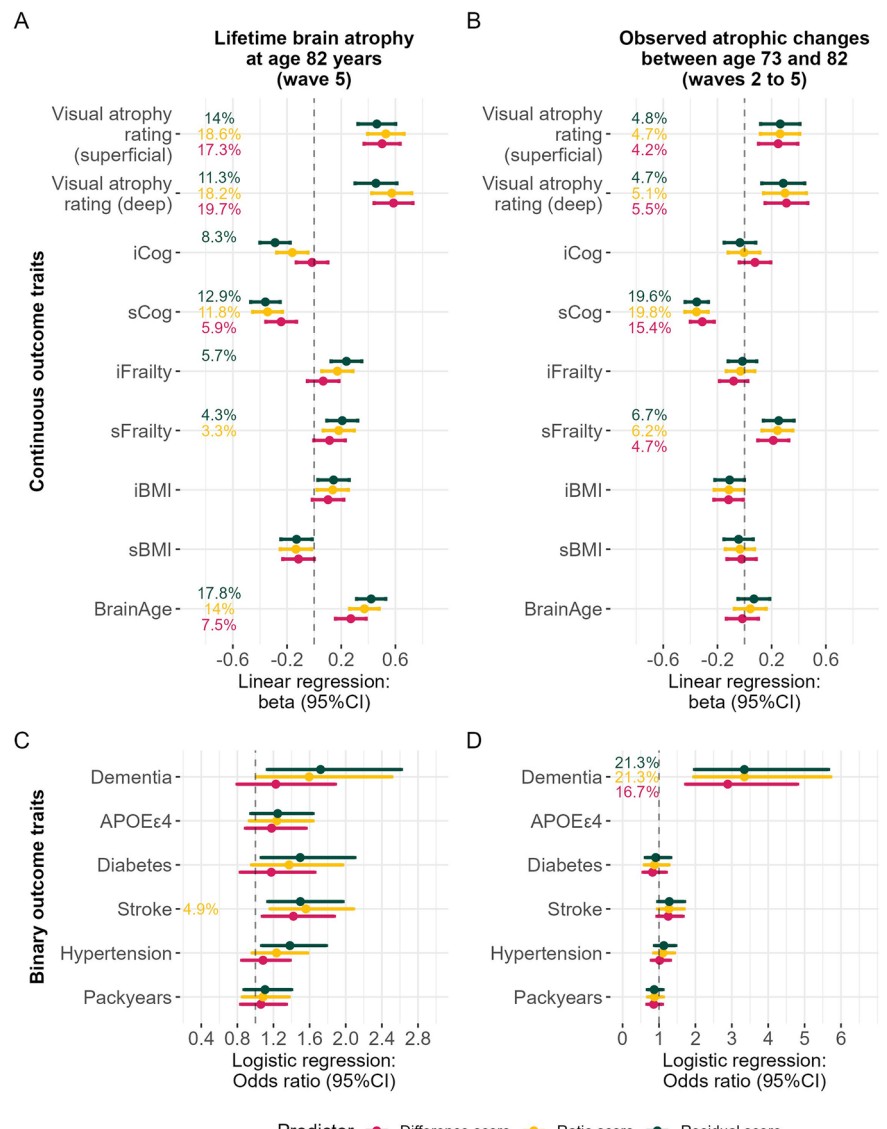

**Fig. 2 | Associations with health-related phenotypes in LBC1936 for lifetime brain atrophy (LBA) and observed atrophic changes ($N = 286$).** No additional covariates were included. Atrophy estimated with the ratio and residual method were flipped to match the difference score whereby larger values represent more brain atrophy. Associations with continuous traits in panels (**A**, **B**) were calculated with linear regressions, where the outcome was always one of the traits listed on the y-axes and the predictor was either LBA or longitudinal atrophic changes. It may be counterintuitive to include *APOEε4* status as an outcome, but we make no claims of causality or directionality and prioritised treating all outcome traits equally. Beta effect sizes indicate change per *SD* in the health trait (e.g., *iCog*). Percentages indicate variance explained ($R^2$) in the health trait, and is only printed if the association is statistically significant ($p < 0.05/15$ traits; two-sided tests). Associations with binary traits in panel (**C**, **D**) were calculated with logistic regressions; $R^2$ estimates were obtained with Nagelkerke's $R^2$. Odds ratios indicate the increased chances of having one of the diseases listed on the y-axis associated with one SD deviation in LBA (or observed atrophic changes). Only *packyears* was analysed with hurdle regression where $R^2$ is inferred with maximum likelihood pseudo $R^2$. Confidence intervals are at 95% and were calculated as beta ± 1.96*SE for continuous traits, and exp(beta ± 1.96*SE) for binary traits. Note when interpreting the results that the x-axes on panel C-D are scaled differently. Variables for which we extracted intercepts and slopes (*Cog, Frailty, BMI*) were relative to the same baseline at age 73 as observed atrophic changes, but LBA represents loss since maximum brain size many years earlier. Visually rated atrophy was assessed at age 76 (wave 3) which was the last available time point. Ideally, we would have preferred to include visually rated atrophy at age 82 (wave 5), so that LBA, the final time point of longitudinal atrophic changes, and visually rated atrophy would have been recorded at the same time point.

phenotype might index longer-term changes of ageing that are not specifically disease-related and therefore less precisely distinguish participants with dementia diagnoses.

In the UKB sample, LBA was significantly associated with greater general cognitive function (*g*-factor; *beta* $_{residual} = 0.23$ [95% CI = 0.20–0.26]), older brain age gap (*beta* $_{residual} = 0.27$ [95% CI = 0.24–0.30]), more severe frailty (*OR* $_{residual} = 1.24$ [95%CI = 1.16–1.32]), increased likelihood to have diabetes (*OR* $_{residual} = 1.51$ [95% CI = 1.33–1.71]), and hypertension (*OR* $_{residual} = 1.46$ [95%CI = 1.37–1.56];

Fig. S7). Associations were significant despite the relatively healthy nature of UKB neuroimaging participants: raw LBA estimates suggested UKB participants were similarly healthy to those in the MRi-Share and HCP cohorts who were chronologically four decades younger, on average (Figs. S1, 2). This may be a reflection of healthy volunteer selection biases[27,28], or different approaches to MRI processing. Associations were largely consistent in males and females separately, but the association between LBA and diabetes was driven by males (Fig. S8).

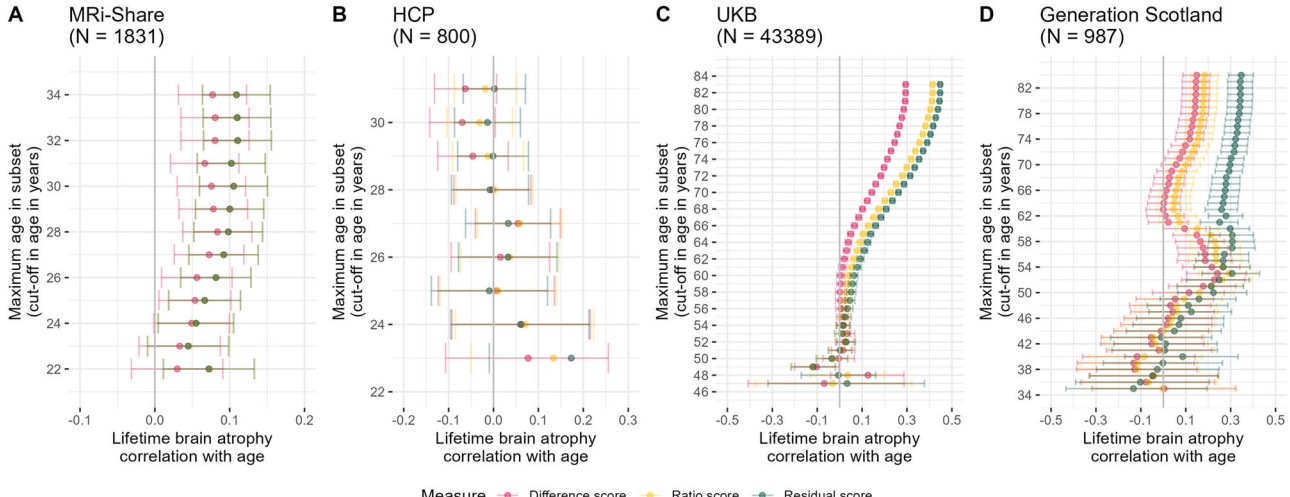

**Fig. 3 | Lifetime brain atrophy (LBA) is moderated by sample age across three out of four earlier- and later-life cohorts.** Age moderation of LBA estimated with the difference, ratio, and residual method. The *y*-axes indicate the maximum age in sample subsets taken to calculate the respective plotted age correlations. For example, a value of 35 on the *y*-axis means that the sample subset used to calculate the age correlation contains participants 35 years or younger. Error bars were calculated as $r \pm 1.96 \cdot SE$. The ratio and residual scores have been flipped (i.e., multiplied by -1) so a larger value corresponds to more brain atrophy. Panel A displays age correlations in the MRi-Share cohort ($N = 1831$; age range 18–34 years), Panel B the Human Connectome Project ($N = 800$; age range 22–31 years), Panel C the UKB

($N = 43{,}389$; age range = 45-83), and Panel D the Generation Scotland sample ($N = 987$; age range 26–84). The kink in the Generation Scotland diagram is likely a reflection of the bimodal distribution of the LBA phenotypes, which least strongly affected the residual score (Fig. S3). This kink was likely driven by the males in the sample (Fig. S19). Note that the mismatch in the reported UKB sample size in this figure and Supplementary Data 6 is due to this figure requiring one MRI scan only which is available from many more participants than two repeated measures as required by analyses of longitudinally-observed atrophic changes. Age correlations were non-significant when repeated in an unrelated HCP sample ($n = 326$; Fig. S20).

## Measures of LBA indicate age-associated brain shrinkage

When considering only one single time-point, LBA estimated with all three methods was substantially correlated with age in the UKB, which is a mid-to-late life age-heterogeneous sample (46–81 years). We show that the strength of this correlation is moderated by sample age. That is, we created subsamples of successively older upper-age limits within a cohort, which caused the correlation between age and LBA to increase towards older adulthood (Fig. 3C). For example, LBA $_{residual}$ was correlated with age at $r = 0.45$ [95%CI 0.44–0.46] in the full UKB sample ($N = 43{,}389$; age range = 45–81 years). However, considering a subsample aged <62 years produced an age correlation of $r = 0.1$ [95% CI 0.08–0.11] ($N = 15{,}461$), and the same correlation was zero [95%CI −0.001–0.05] when considering only below 55-years-olds ($N = 5475$). Results remained stable but yielded fewer significant age correlations when this analysis was repeated for males and females separately (Fig. S19).

This same trend was evident in the young-to-old-age Generation Scotland sample ($N = 987$; age range 26–84 years; Fig. 3D) in that age correlations were strongest including older participants. This trend even existed in young adults, namely the MRi-Share cohort including mainly university students (Fig. 3A, $N = 1831$; age range 18–34 years), but not the HCP cohort (Fig. 3B, $N = 800$; age range 22–31 years). Note that the raw HCP data showed strong age correlations for ICV (unexpectedly small skulls in older females contradicting our pre-registered expectations; $r = -0.20$, $p = 4.2 \times 10^{-11}$; Supplementary Material; 'Deviations from pre-registration'). Following a pre-registered age cut-off, we excluded 303 participants >31 years in the HCP data that we analysed (e.g., in Fig. 3B). Integrating TBV and ICV directly in the model [$lm(age \sim TBV + ICV)$] considerably increased associations with age compared with a two-step residual approach [$LBA\ residual = resid(lm(TBV + ICV));\ lm(age \sim LBA_{residual})$], especially in the UKB and at smaller older-age subsamples. Large variance inflation factor values above 5 suggest that the one-step approach suffers from variance inflation (SFig. 41). Additional analyses in the age-homogeneous and repeatedly MRI-scanned LBC1936 cohort showed that LBA increased with advancing chronological age, even when

using MRI measures that were processed independent of scans from other visits (i.e., in this analysis MRI data was processed with the FreeSurfer cross-sectional rather than the longitudinal processing stream; Fig. S18).

## LBA moderately captures longitudinally-observed atrophic changes over 9 years

To quantify the extent to which LBA approximated longitudinally-observed atrophic changes, we derived atrophy scores from both cross-sectional MRI data measured at a second of two occasions (i.e., LBA), and from two longitudinal MRI scans measured 9 years apart in LBC1936. We term the latter *observed atrophic changes*, capturing a time window shorter than, but overlapping with the one captured by *lifetime* BA (Fig. 1). For all computational methods, the Pearson's correlation was modest (LBC1936; $N = 277$): $r$ $_{residual} = 0.36$ [95% CI = 0.25–0.46], $r_{ratio} = 0.29$ [95% CI = 0.18–0.40], $r_{difference} = 0.30$ [95% CI = 0.19–0.40]; Fig. S4; very similar sex-split correlations in Supplementary Data 7. This suggests that LBA moderately captured variation indicated by longitudinal atrophic changes, despite the fact that the two measures cover time windows of different lengths (i.e., lifetime vs. 9 years in LBC1936). The correlations appeared substantial given that LBC1936 participants demonstrated, on average, limited atrophic changes across the measured 9 year window (mean $TBV_{time\ 1} = 1028\ mm^3$, mean $TBV_{time\ 2} = 959\ mm^3$; Fig. S16, 17) which likely limited the potential for a strong correlation between LBA and observed atrophic changes.

The UKB data confirmed that LBA moderately captured longitudinal atrophic changes: their correlations were numerically lower than in LBC1936, but still substantial considering atrophic changes in UKB only covered a 4 year time window ($N = 4674$; $r$ $_{residual} = 0.29$ [95% CI = 0.27–0.32], $r_{ratio} = 0.24$ [95% CI = 0.22–0.27], $r_{difference} = 0.21$ [95% CI = 0.18–0.24]; Fig. S5; very similar sex-split correlations in Supplementary Data 7. Given the shorter 4 year time window—likely impacting measurement reliability[29]—mean atrophic changes were even more limited in UKB (mean $TBV_{time\ 1}$ at age 73 = 1185 $mm^3$, mean $TBV_{time\ 2}$ at age 82 = 1171 $mm^3$; Fig. S14, 15) than they were in LBC1936.

**Table 1 | Genetic characteristics of lifetime brain atrophy (LBA)**

| Traits | GCTA SNP-heritability (SE) | Mean $\chi^{2a}$ | LDSC intercept (SE) | LDSC heritability (SE) | Independent significant[b] GWAS hits | GWAS loci |
|---|---|---|---|---|---|---|
| LBA (residual score) | 0.41 (0.01) | 1.23 | 1.01 (0.008) | 0.26 (0.02) | 93 | 28 |
| LBA (ratio score) | 0.42 (0.01) | 1.24 | 1.02 (0.009) | 0.27 (0.02) | 112 | 30 |
| LBA (difference score) | 0.47 (0.01) | 1.27 | 1.02 (0.010) | 0.30 (0.02) | 128 | 37 |
| Cerebrospinal fluid (CSF) volume | 0.39 (0.01) | 1.24 | 1.01 (0.009) | 0.27 (0.02) | 48 | 21 |
| Total brain volume (TBV) | 0.56 (0.01) | 1.34 | 1.03 (0.011) | 0.36 (0.03) | 158 | 56 |
| Intracranial volume (ICV) | 0.58 (0.01) | 1.35 | 1.03 (0.011) | 0.37 (0.03) | 138 | 45 |

[a]Mean χ2 calculated in HapMap3 SNPs; [b]Genome-wide significance threshold (two-sided association test calculated with REGENIE) = $p < 5 \times 10^{-8}$, independence determined at $r^2 < 0.6$ calculated from UKB release2b 10 k European reference panel. Abbreviations: *GCTA* Genome-wide complex trait analysis[30], *SNP* single-nucleotide polymorphism. *LDSC* linkage disequilibrium score regression[31], GWAS genome-wide association study.

Correlations were very similar when using the T1-scaling factor instead of ICV to estimate LBA $_{residual}$ (Fig. S6). Correlations across all cross-sectional and longitudinal measures in LBC1936 and UKB are displayed in Figs. S4, 5, additionally illustrating that FreeSurfer-based CSF volume does not correspond to LBA likely because it only captures the ventricle system (not the subarachnoid space; Methods; 'MRI processing robustness checks in LBC1936'). Our pre-registered efforts (https://osf.io/gydmw/) to artificially equate timelines for LBA and longitudinal atrophic changes were not safely interpretable due to the relatively healthy nature of UKB participant's brains compared to the much younger cohorts MRi-Share and HCP (Figs. S1, 2; Supplementary Material; 'Sample specificity of LBA norms and the role of ICV').

Analyses across younger and older cohorts indicated that TBV and ICV were more strongly correlated (i.e., more similar measures) in the younger cohorts: $r = 0.96$ [95%CI 0.95–0.96] in MRi-Share [age: M (range) = 22 (18–35) years]; $r = 0.92$ [95%CI 0.90–0.93] in HCP [age: M (range) = 27 (22–31) years], as compared to the older cohorts: $r = 0.90$ [95%CI 0.89–0.90] in UKB [age: M (range) = 62 (46–82) years]; $r = 0.77$ [95%CI 0.74–0.80] in Generation Scotland [age: M (range) = 62 (26–84) years]; $r = 0.81$ [95%CI 0.78–0.83] in LBC1936. Although confidence intervals overlap for estimates from different cohorts, this was consistent with our pre-registered expectations and is compatible with the assumption of our study that ICV is a measure of prior brain size whereby the brain approximately fills the entire intracranial vault in young adulthood – prior to the occurrence of any atrophy that may be detectable in older adulthood.

**Genome-wide association study of LBA**

The second part of this paper explores the genetic bases of LBA by reporting its single nucleotide polymorphism (SNP)-heritability, and calculating its genome-wide associations in the UKB ($N = 43,110$) and genetic overlap with other structural neuroimaging and neurodegenerative traits.

**SNP heritability.** SNP-heritability was calculated using Genome-wide Complex Trait Analysis (GCTA v1.94.1)[30] in European-only UKB genotype data with a cryptic relatedness cut-off at 0.025 ($N = 38,624$; Table 1). The SNP-heritability of the residual score was significantly lower (41% [95% CI = 38–43%]) than for either the ratio (42% [95% CI = 40–44%]) or difference score (47% [95% CI = 45–49%]), when testing their differences with a conservative z-test accounting for their genetic correlations (LBA $_{difference}$: $z = 3.54$, $p = 0.0002$; LBA $_{residual}$: $z = 8.02$, $p = 5.38 \times 10^{-16}$). It is notable that the residual score is, by construction, entirely independent of ICV (*Box S1*) – which has an even higher heritability of 58% [95% CI = 55–61%]. That is, even when the residual score was derived from the residuals of two highly correlated and highly heritable phenotypes, it carried substantial systematic genetic signal (*amplifier effect*). It supports our phenotypic observations that LBA $_{difference}$ is a less specific, broader

phenotype than LBA $_{residual}$ that it yielded larger heritability estimates (47% vs. 41%), and more GWAS loci (37 vs. 28 loci).

**GWAS associations.** We performed European-only GWAS analysis in REGENIE (v.3.4)[32]. Pre-registered covariates included age, sex, acquisition site and time, scanner positions, 40 genetic principal components (PCs), genotyping array and genotyping batch. FUMA v1.5.2[33] (default settings) identified 28 independent genomic risk loci associated with LBA $_{residual}$ (Fig. 4; Supplementary Data 10–15). The most significant SNP, *rs142005327* ($p = 3.16 \times 10^{-46}$) was part of a locus spanning from 120777961 to 121033191 on chromosome 7. Gene prioritisation in FUMA using expression quantitative trait loci (*eQTL*) mapping (PsychENCODE[35], BRAINEAC[36], GTEx v8 Brain[37]) indicated that our most significant risk locus contained SNPs that are known *eQTLs* of the *WNT16* gene expressed in the brain. The *WNT* gene family encodes signalling proteins involved in oncogenesis and developmental processes including regulation of cell fate[38]. According to the GWAS Catalogue, SNP *rs142005327* was previously linked with brain morphometry[39,40], brain age[41], but also bone mineral density[42] and osteoporosis[43] which are both age-related diseases and were linked to *WNT* signalling. Both *APOE* SNPs *rs7412* and *rs429358* were not associated with either of the LBA phenotypes (LBA $_{residual}$; $p$ $_{rs7412} = 0.431$; $p$ $_{rs429358} = 0.735$).

LBA $_{ratio}$ identified 30 genomic risk loci, 23 of which overlapped with the 28 genomic loci identified by LBA $_{residual}$ in their genomic locations (Supplementary Data 12). LBA $_{difference\ score}$ identified 37 genomic risk loci, 17 of which overlapped with the 28 LBA $_{residual}$ genomic loci (Supplementary Data 14). This suggests LBA $_{ratio}$ and LBA $_{difference}$ captured broader and less specific genetic signal compared to LBA $_{residual}$. The top hit in LBA $_{ratio}$ was *rs142005327* and the top hit in LBA $_{difference}$ was *rs10668066*, both located in chromosome 7. The nearest gene for both these SNPs was the *WNT16* gene. All three LBA phenotypes produced similar looking Manhattan plots (Figs. S21, 22). 71% of the independent significant SNPs and 90% of the mapped genes identified for LBA $_{residual}$ (Supplementary Data 11) were also captured by LBA $_{ratio}$ (Supplementary Data 13). 34% of the independent significant SNPs and 50% of the mapped genes identified for LBA $_{residual}$ were also captured by LBA $_{difference}$ (Supplementary Data 15). Manhattan plots for TBV, ICV, and CSF volume calculated in the same sample as the LBA phenotypes are in Fig. S23–25. SNP-by-age interaction analyses indicated no evidence for significant SNP-by-age interactions on any of the LBA phenotypes ($p < 5 \times 10^{-8}$; Fig. S26–37; Supplementary Data 8), which included both *APOE* SNPs *rs7412* and *rs429358* (LBA $_{residual}$: $p$ $_{rs7412} = 0.195$; $p$ $_{rs429358} = 0.660$; detailed outline in *SMaterials*).

Non-pre-registered investigations using the tool by Privé, Luu[44] delivered empirical evidence that only 8 genetic PCs should be included for this UKB neuroimaging subset (*SMaterials*; *GWAS re-analysis adjusting for 8 genetic PCs only*). Adjusting for 8 instead of

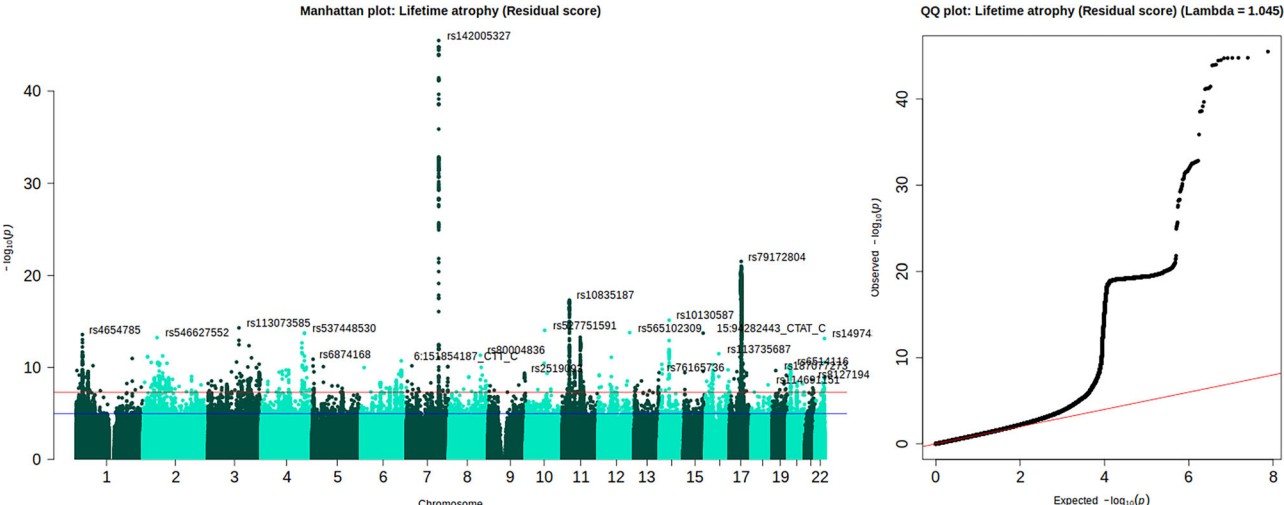

**Fig. 4 | Genome-wide association study of lifetime brain atrophy (LBA).** Left: Manhattan plot for LBA inferred with the residual method indicating top GWAS hits (minor allele frequency > 0.01, imputation INFO metric > 0.8) in 28 genomic loci ($N = 43,110$). Right: Quantile-quantile (QQ) plot indicating deviation of polygenic signal from the expected null signal on the red line. The genomic inflation factor was 1.045. Both panels were plotted with the *qqman* package in R[34]. Further evaluation and visualisation of the QQ plot in Fig. S38 shows that the step in the QQ-plot shown here disappeared when considering HapMap3 SNPs only.

40 genetic PCs did not affect LBA $_{residual}$, but LBA $_{difference}$ and LBA $_{ratio}$ captured more significant GWAS loci (Supplementary Data 9), likely reflecting increased power by only including PCs that showed no evidence for capturing complex LD structure. Integrating TBV and ICV directly in the GWAS model [$lm(SNP \sim TBV + ICV)$] produced very similar results to the two-step residual approach [$lm(SNP \sim LBA_{residual})$]: mean $\chi^2 = 1.23$; LDSC intercept (SE) = 1.01 (0.008); LDSC heritability (SE) = 0.26 (0.02); independent significant SNPs = 88; significant GWAS loci = 27. GWAS derived from both approaches were highly genetically correlated at 0.97 (0.07); the genetic correlation was non-significant between ICV and the adjustment method GWAS ($r_g = 0.046$; $SE = 0.05$).

**Genetic correlations.** To characterise LBA, we calculated genetic correlations using bivariate LDSC[45] (MAF > 0.01, INFO > 0.9) with different structural neuroimaging traits (Fig. 5). LBA $_{residual}$ had a moderate negative genetic correlation with our TBV GWAS ($r_{g\ residual} = -0.37$; $SE = 0.05$) and was uncorrelated with the TBV GWAS by Zhao, Luo[46] ($r_{g\ residual} = -0.05$; $SE = 0.05$), both of which did *not* include ICV as a covariate. The same measure of LBA $_{residual}$ was strongly associated with the TBV GWAS by Smith, Douaud[47] ($r_{g\ residual} = -0.81$; $SE = 0.06$) where the TBV-associated SNP effects in Smith et al.[47] were adjusted for a close proxy of ICV (i.e., T1 scaling factor) as a covariate (negative $r_g$ because direction of effects flipped for LBA $_{residual}$ but not for Smith et al. GWAS[47]). The large magnitude of this genetic correlation suggests that the ICV-adjustment in Smith et al.[47] mirrors our definition of LBA $_{residual}$, indicating that this T1 scaling factor-corrected phenotype connotes genetic correlates of brain change rather than total brain volume alone, which the name would suggest.

Some genetic correlations underline the desirable properties of the residual method compared to the difference or ratio method ($r_g$ across three LBA scores = 0.72–0.96). Only LBA $_{residual}$ was non-significantly genetically correlated with ICV ($r_g = -0.09$; $SE = 0.05$). The other computational methods produced larger and significant genetic correlations with ICV ($r_{g\ ratio} = 0.35$, $SE = 0.05$; $r_{g\ difference} = 0.75$; $SE = 0.06$), which confirms phenotypic observations that only the residual method captures variance un-contaminated by ICV baseline levels. Furthermore, the difference method appears to have induced the counter-intuitive genetic correlation between greater LBA and larger TBV ($r_g = 0.37$; $SE = 0.04$); as opposed to the opposite direction of effects indicated by the ratio ($r_g = -0.11$; $SE = 0.04$) and the residual method ($r_g = -0.37$; $SE = 0.05$). We speculate that a negative genetic correlation between LBA and TBV would be theoretically more intuitive where a simple measure of TBV captures, among other factors, both early-life neurodevelopment as well as later-life neurodegeneration (i.e., decline which is compatible with a negative direction of effects). However, this would be less intuitive if neurodevelopment makes a substantially larger contribution to later-life differences in TBV than ageing-related neurodegenerative processes that likely occur over many decades. A GWAS-by-subtraction model[48] confirmed that the genome-wide TBV-associated residuals of ICV near perfectly genetically overlapped with LBA $_{residual}$ ($r_g = 0.95$ between LBA $_{residual}$ and TBV residualised for ICV on a genome-wide level). LDSC-heritability in males (LBA $_{difference} = 0.37$ [0.03], LBA $_{ratio} = 0.30$ [0.03], LBA $_{residual} = 0.27$ [0.03]) was similar but numerically larger than heritability in females (LBA $_{difference} = 0.27$ [0.03], LBA $_{ratio} = 0.24$ [0.03], LBA $_{residual} = 0.25$ [0.03]). Genetic correlations across LBA phenotypes calculated in males ($n = 20,453$) and females ($n = 22,657$) separately were large ($r_g = 0.91$–1.00; Fig. S39) indicating that genetic signal in the GWAS including all participants was largely unrelated to sex.

We also calculated genetic correlations between LBA and an array of neurogenerative diseases for which we identified well-powered GWAS summary statistics. Alzheimer disease[49] and related dementias[50], and amyotrophic lateral sclerosis (ALS)[51] were genetically uncorrelated with LBA (Fig. 5). Parkinson disease[52] yielded significant genetic correlations with estimates of LBA $_{difference\ \&\ ratio}$, but not LBA $_{residual}$. Because the difference and ratio scores substantially captured ICV-associated genetic variance ($r_{g\ LBA\ ratio\ \&\ ICV} = 0.35$, $SE = 0.04$; $r_{g\ LBA\ difference\ \&\ ICV} = 0.75$, $SE = 0.05$) and the residual score did not ($r_{g\ LBA\ residual\ \&\ ICV} = -0.09$, $SE = 0.05$; ns.), we suggest that the significant associations between Parkinson disease and the difference and ratio scores were likely driven by ICV baseline differences ($r_{g\ Parkinson\ disease\ \&\ ICV} = 0.19$; $SE = 0.05$) rather than brain changes[53].

Our analyses do not include a genetic correlation with longitudinal atrophic changes from Brouwer, Klein[8] —the largest multicohort GWAS of longitudinal lifespan brain changes ($N \sim 16,000$)— because this set of GWAS summary statistics produced a negative LDSC heritability estimate. This means we were unable to detect systematic polygenic signal across a subset of HapMap3 SNPs, despite this

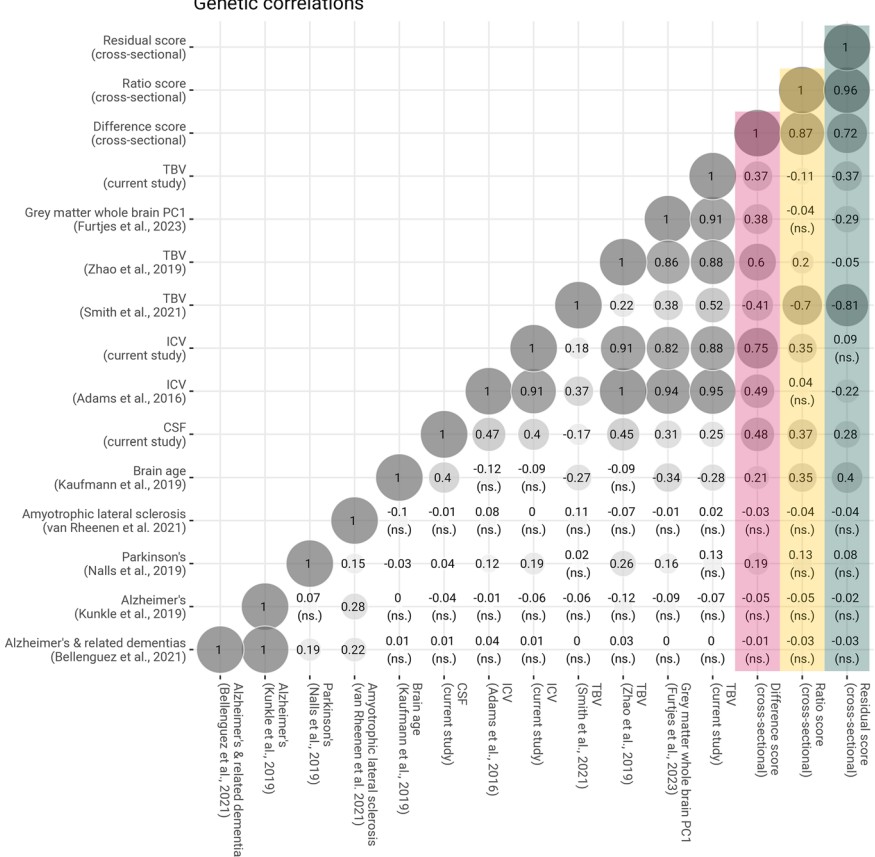

**Fig. 5 | Genetic correlations inferred via linkage disequilibrium score regression (LDSC) between lifetime brain atrophy (LBA) calculated by three computational approaches (residual, ratio and difference score; $N = 43,110$) and other structural neuroimaging and neurodegenerative phenotypes.** Genetic correlations with the difference score are coloured pink, those with the ratio score are coloured yellow, and those with the residual score are coloured green. The notion 'cross-sectional' in the figure indicates that this measure was calculated based on a single, cross-sectional MRI scan. To match procedures in phenotypic investigations, genetic correlates associated with LBA inferred with the ratio and residual method were flipped (i.e., multiplied by -1) whereby positive effect sizes indicate single-nucleotide polymorphisms (SNPs) that increase the risk of greater LBA, which is the same direction of effects tagged by LBA inferred with the difference method. Unless marked with *ns.* genetic correlations were statistically significantly from zero at 99% confidence intervals (critical z score = 2.807). Abbreviations: TBV total brain volume, ICV intracranial volume, CSF cerebrospinal fluid volume.

trait having been reported to carry small but extant genetic signal (LDSC $h^2$ Z statistic = 1.24 reported in Brouwer et al.[8]). Unlike LBA, Brouwer et al.[8] indicated two top SNPs in chromosomes 13 and 16, and identified no correlates in chromosome 7, or any SNP-correlates associated with *WNT16* signalling.

## Discussion

It was the aim of this study to externally validate and explicitly characterise a cross-sectional measure of LBA using three computational modelling techniques [difference (*ICV-TBV*), ratio (*TBV/ICV*), and regression-residual method (TBV - ICV)], which were then carried forward to conduct the largest GWAS of LBA to date ($N = 43,110$). Our phenotypic investigations support that this straightforward definition of LBA (derived from a single MRI scan) yielded meaningful variance that sensibly correlated with brain atrophy estimated from neuroradiological rating scales, cognitive functioning, and frailty. It was also moderated by chronological sample age, and LBA moderately captured longitudinal atrophic changes measured over a shorter time period (< 9 years). This moderate correlation is noteworthy because it remained robust despite reliability challenges[29], and differences in time windows covered by the measures (i.e., longitudinal atrophic changes cover years, cross-sectional LBA measures cover decades).

Given the much greater availability of samples with a single MRI scan and genetic information, LBA represents a cost-effective GWAS phenotype to boost statistical power for discovery of the molecular genetic bases of lifetime changes in brain atrophy. We demonstrated that these phenotypic properties were maximally captured when LBA was estimated with the residual rather than the difference or ratio method. Estimating LBA $_{residual}$ exploited an amplifier effect whereby it remained statistically independent of neurodevelopmental baseline differences in head size (i.e., uncorrelated with ICV), at the same time as maximally focusing our analyses on age-related neurodegenerative brain volume *changes*. Our study explicitly demonstrates that the TBV-associated residuals adjusted for ICV capture ageing-related variance (in older cohorts) that TBV or ICV variables could not have captured in isolation. This goes beyond simply adjusting for (potentially) confounding effects associated with ICV, as is often given as justification for ICV adjustment[21]. Our FreeSurfer-based LBA $_{residual}$ remained modestly biased towards manually-segmented ICV in LBC1936 ($N = 631$), and future studies should test the extent to which this bias pervades other automated methods.

Building on this carefully described measure of LBA, we performed large-scale genome-wide investigations ($N = 43,110$) to complement previous efforts based on meticulously collected longitudinal MRI scans (N ~ 16,000; [8]). LBA was substantially heritable ($h^2_{SNP\ GCTA} = 41\%$ [95% CI = 38–43%]), and our GWAS summary statistics yielded strong polygenic signal in the LDSC framework ($h^2$ Z statistic = 13). These characteristics qualify our GWAS summary

statistics to form a useful contribution to future investigations of neurodegenerative diseases (e.g., using polygenic scores, or Genomic Structural Equation Models)[54].

The most significant genomic risk locus of LBA indicated the *WNT16* gene in chromosome 7 which has previously been linked to reduced cortical volume in Alzheimer patients[55] and has also been implicated in other age-related traits like brain age[41] and osteoporosis[43]. *WNT* signalling has been linked to neurodegenerative diseases such as Parkinson disease and ALS[56], as well as amyloid-beta-related pathogenesis in Alzheimer disease where *WNT* was suggested to drive synaptic loss[57]. LBA did not demonstrate significant genetic correlations with existing GWAS of neurodegenerative diseases such as Alzheimer and Parkinson disease (or significant genetic correlation between LBA $_{difference}$ and Parkinson we suggest are driven by ICV), but their genetic correlation with LBA may be subject to insufficiently powered neurodegenerative sample sizes. Future, more detailed genetic follow-up analyses (for example, performing gene prioritisation or Mendelian Randomization) should be conducted in larger, better-powered, and multi-ancestral investigations. This paper is a precursor for a larger consortium-level meta-analytic GWAS of this single-time-point definition of LBA which promises much larger sample sizes.

The work presented here highlights the importance of carefully considering and interpreting different methodological approaches used to estimate lifetime changes (i.e., difference, ratio, residual method) which includes adjusting for the appropriate covariates in the GWAS analysis. Conceptually, the different LBA estimates make different underlying assumptions that influence their biological interpretation. The difference score reflects an absolute measure of total volume loss from an inferred baseline, which does not account for individual differences in initial adult size which may be influenced by sex and neurodevelopmental factors. This contrasts with relative measures (though the reference group remains an important consideration). As we show, only atrophy calculated with the residual method will be phenotypically and genetically uncorrelated with ICV baseline differences, and atrophy calculated using the difference or ratio method highly likely comprises results driven by ICV baseline differences. Hence, the difference and ratio methods tag genetic signal that amalgamates late-life neurodegeneration (or any longitudinal changes), as well as developmental processes indicated by ICV ($r_g$ between ratio score and ICV = 0.35; $r_g$ between difference score and ICV = 0.75). Moreover, the phenotypic and genetic results are in keeping with the literature on brain and cognitive reserve[58]. Though all three LBA measures reflect some aspect of brain reserve (the raw quantity of cerebral resource that allows a given brain to better tolerate neuropathology), their correlations with ICV indicate an important distinction between 'static' and 'dynamic' reserve[59]. LBA measures that are more strongly correlated with ICV (e.g., for LBA $_{difference}$) are more reflective of pre-existing brain and head size differences from when they were maximal ('static reserve'). By contrast, zero-correlations with ICV (e.g., for LBA $_{residual}$) can be considered a marker of an individual's dynamic brain reserve at the time of the scan.

This consideration also applies to interpreting GWAS results from previous studies of TBV, where adjusting for ICV re-directed the tagged variance away from broad differences in brain size, towards lifetime neurodegenerative changes more specifically. For example, consider the *big40* TBV GWAS by Smith, Douaud[47] who followed the well-established practice of adjusting GWAS analyses for a close proxy variable of ICV (i.e., T1 scaling factor). This ICV-adjusted TBV GWAS was strongly genetically correlated with LBA $_{residual}$ ($r_g = -0.81$) which we postulate should capture change, and had much weaker genetic correlations with other TBV GWAS that were unadjusted for ICV ($r_g = 0.22-0.52$)[46] (and current study)[60]. By contrast, the unadjusted TBV GWAS yielded much smaller genetic correlations with LBA $_{residual}$ ($r_g = -0.05 - -0.37$) because unadjusted TBV tags a broader and less specific phenotype underpinned by both early-life neurodevelopment

and late-life neurodegeneration. This reasoning only holds in later-life cohorts, and does not extend to young, pre-neurodegenerative cohorts where TBV is much more similar to ICV. LBA would hence not index neurodegeneration but instead healthy variability in earlier life development which is not the focus of this study. Our work highlights the requirement for transparent characterisations, not only of the GWAS phenotype, but also of the residuals that remain after adjusting for covariates.

This study has several limitations. We focused on LBA across the whole brain—rather than individual regions or tissue types – because precision and accuracy of a volume estimated in MRI is maximal, the larger the volume[61]. The disadvantage to maximising measurement reliability is that LBA is a very broad phenotype biologically underpinned by a collection of many different tissue types and biological mechanisms. For example, it is unclear to what extent LBA captures cellular disintegration within brain regions (i.e., grey matter), or between brain regions (i.e., white matter). Gross morphometry from structural MRI does not indicate whether neurons, microglia, astrocytes, or other cell types are affected, or whether neurodegeneration resulted from ischaemic damage, though this applies to regional as well as global MRI measurements. It would also not allow the detection of specific signatures of frontal and anterior temporal atrophy observed in fronto-temporal dementia[62], or other regional neurodegenerative patterns in other diseases. Future studies are required to develop reliable ways to record those more specific and fine-grained changes and how they may relate to the genetics of brain-wide atrophy analysed here. The reliability of longitudinal estimates was likely limited, especially in the UKB where follow-up intervals were short[29,63]. Our residual scores were constructed with reference to full samples (or males and females in sex-split analyses) where selecting appropriate age ranges was shown to influence results[21]. Future work is needed to assess the impact of using different reference samples. Our use of linear statistical methods implies linear processes, which we argue is a reasonable assumption for the short time windows each cohort covers, but this may not be correct for lifespan studies. Analyses in LBC1936 confirm that LBA extracted from different FreeSurfer versions capture highly similar variance ($N = 581$; Methods; 'MRI processing robustness checks in LBC1936'). While manual ratings and tracings are also subject to forms of error and bias, the above interpretation must consider that FreeSurfer-based LBA measures show modest bias towards larger manually segmented ICV (shown in LBC1936 $N = 631$; Methods; 'MRI processing robustness checks in LBC1936'), a challenge that affects most large-scale MRI studies including our genetic findings in UKB.

To conclude, this study provided a comprehensive phenotypic characterisation of LBA estimated from a single MRI scan, using five cohorts across adulthood, which allowed boosting statistical power for better genetic discovery. LBA was substantially heritable, had strong polygenic signal, and the strongest genetic correlates highlighted the *WNT16* pathway as one potential mechanism underpinning broad, late-life neurodegeneration. Our work underlined the importance of using appropriate computational approaches, where we demonstrated that only the residual method can focus analyses of TBV towards late-life neurodegeneration independent of earlier-life neurodevelopmental processes. Finally, we showed that it is imperative for neuroimaging genetics studies to transparently characterise not only their GWAS phenotype, but also the remaining residuals after covariate adjustment because the inclusion of ICV (or a close proxy variable such as the T1 scaling factor) as a covariate has important implications for the interpretation of phenotypic and genetic results.

## Methods
### Sample descriptions
All cohorts listed below were analysed due to their available T1 neuroimaging data. This collection of cohorts of differing mean cohort ages allowed us to consider young, middle- and older-aged adults. All

cohorts followed appropriate ethical regulations and obtained informed consent from their participants. The board that approved the study protocol is listed below as part of each study description. We excluded participants whose TBV was larger than ICV, whose TBV or ICV estimate were zero, as well as outliers outside of 10SDs. There were no exclusions if not mentioned in the cohort descriptions below.

## Young adults

**Human Connectome Project (HCP).** The HCP is a cohort of healthy young adults (N = 1113; age range 22–35) that were recruited from ongoing studies as part of the Missouri Family Study[64]. HCP was approved by the Washington University in St. Louis Institutional Review Board. Data was collected between 2010 and 2015. Exclusion criteria mainly included severe diseases (e.g., epilepsy, multiple sclerosis, cerebral palsy), as well as criteria that would have prohibited safe MRI scanning (e.g., metal or devices in the body, claustrophobia). The HCP analysis team performed MRI data processing[65,66] in Free-Surfer (v5.2)[67], from which we downloaded ICV (FS label: *eTIV*) and TBV (FS label: *BrainSegNotVent*) estimates. To maximise sample size, our analyses included related individuals, but were also repeated in unrelated individuals (randomly sampled from *Family_ID* variable). We removed six participants from the sample because their TBV estimate was larger than their ICV estimate (the brain cannot be larger than the skull). Refer to an explanation for why only below 31 year-olds were included in the analysis in the '*Deviations from the pre-registration*' section in *Supplementary Materials*. Final distributions of TBV, ICV, and the LBA scores in 800 participants (393 females) are displayed in SFig. 3. Access to restricted data was obtained from the Washington University School of Medicine.

**MRi-Share.** The MRi-Share cohort consists of 1831 university students in Bordeaux, France, who were recruited as volunteers from the larger online i-Share study (www.i-share.fr). MRi-Share data was approved by the French National Commission on Informatics and Liberty (Commission Nationale de l'Informatics et des Libertés; CNIL). Data was collected between 2013 and 2017. Participants were aged between 18 and 35 years, and exclusion criteria were pregnancy, and characteristics that would have prohibited safe MRI scanning (e.g., claustrophobia). The MRI acquisition protocol was designed to match MRI acquisition in the UK Biobank sample (https://www.ukbiobank.ac.uk/). ICV (FS label: *eTIV*) was extracted using FreeSurfer v6.0 (http://surfer.nmr.mgh.harvard.edu/). Structural T1 and FLAIR images were processed by Tsuchida, Laurent[68], where white and grey matter volume were extracted using SPM12 (https://www.fil.ion.ucl.ac.uk/spm/). We calculated TBV by summing white and grey matter volume. Imaging-derived phenotypes were freely available for download online[69]. Final distributions of TBV, ICV, and the LBA scores in 1831 participants (1320 females) are displayed in SFig. 3.

## Middle- and older-aged adults

**UK Biobank (UKB).** The UKB is a prospective population-based cohort including half a million participants in the United Kingdom (https://www.ukbiobank.ac.uk/). The study was approved by the North West Multi-Centre Research Ethics Committee (MREC). Data collection started in 2006 and is ongoing to this date. Participants were recruited as volunteers to collect health-related information from physical measurements, biological samples, and questionnaires during a planned 20 year follow-up[70]. Brain MRI data was collected on a subsample of ~50,000 participants ('time 1')[71], which has so far been repeated for ~4500 of those participants, about 4 years after the initial imaging visit ('time 2')[72]. T1 and FLAIR images were collected across three sites with identical hardware and software, and they were processed cross-sectionally on behalf of the UKB (FreeSurfer v6.0)[73]. TBV and ICV phenotypes are available for tabulated download (field IDs: TBV = 26515, i.e., FS label = *BrainSegNotVent*; ICV = 26521, i.e., FS label = *eTIV*;

T1 volumetric scaling factor = 25000; CSF = 26527)[74]. Data access was granted though application 10279. We removed participants with larger TBV estimates than ICV estimates (n = 17), and another participant because their TBV estimate was nearly five times larger than the average sample TBV. Two extreme participants (outside of 10SDs) were removed as their TBV values were only half their ICV values, and another two participants were removed because their CSF value was larger than their ICV value. Final distributions of TBV, ICV, and the LBA scores in 43,110 participants (22,789 females) are displayed in SFig. 3.

We had pre-registered to re-process MRI data where two time points were available using the FreeSurfer longitudinal processing stream[75]. This, however, was not possible because files provided in field ID 20263 were incomplete for most initial neuroimaging visits (available in ~800 participants), which would have been necessary to run the FreeSurfer longitudinal processing stream. We were unable to obtain complete data from field ID 2025 because the UKB recently changed data download permissions (as part of moving all analyses to their Research Analysis Platform). Instead, we inferred longitudinal changes from TBV estimates produced by the FreeSurfer cross-sectional processing stream as was previously done in Di Biase, Tian[76]. In line with how we treated the cross-sectional data, we also applied a 10SDs cut-off to these repeated measures which removed 6 participants (N $_{total}$ = 4674). Final distributions of the atrophy scores derived from longitudinal data ('longitudinally-observed atrophic changes') are displayed in SFig. 15.

**Generation Scotland.** Generation Scotland is a population-based cohort originating from the Generation Scotland Scottish Family Health Study. The study was approved by the NHS Tayside Research Ethics Committee. Data collection started in 2006 and is ongoing to this date. The neuroimaging subsample of Generation Scotland was collected with the aim to subtype depression using clinical, cognitive, genetic, and brain imaging assessments[77]. MRI data was processed using the cross-sectional FreeSurfer stream for 1043 participants (v5.3; eTIV for ICV estimate). We removed participants with estimates of zero $mm^3$ for TBV or ICV (n $_{removed}$ = 11), and where ICV estimates were smaller than TBV estimates (n $_{removed}$ = 45), resulting in a sample of 987 participants (ages = 26–84 years). Final distributions of TBV, ICV, and the LBA scores in 987 participants (573 females) are displayed in SFig. 3 and their interpretation should consider that Generation Scotland includes both depression cases and controls.

## Older age adults

**Lothian Birth Cohort 1936 (LBC1936).** The LBC1936 is a cohort of community-dwelling older adults, born in 1936, who have been prospectively phenotyped over the past 20 years[78,79]. The study was approved by the NHS Lothian Research Ethics Committee. Most participants had taken part in the Scottish Mental Survey in 1947. Data collection started in 2006 and is ongoing to this date. They were recruited from the Edinburgh City and surrounding Lothians' area. LBC1936 data collection included cognitive, psychosocial and biological measures where we used assessments from 5 waves each acquired 3 years apart during an approximate duration of 9 years. Brain MRI data was acquired across 4 time points (ages 70–87 years wave 2 N = 629, wave 3 N = 428, wave 4 N = 319, wave 5 N = 304)[79], but to match the UKB with only two available time points, we consider wave 2 (first neuroimaging visit) as time 1, and wave 5 (fourth neuroimaging visit) as time 2. MRI data was processed and cleaned by the LBC analysis team (FreeSurfer v5.1.0)[80], and includes variables for ICV (FS label = *eTIV*), TBV (FS label: *BrainSegNotVent*), and CSF volume (field ID: csf_mm3_wX). Estimates of lifetime atrophy were inferred from MRI data processed with the FreeSurfer cross-sectional stream, and estimates of atrophic changes were inferred from MRI data processed with the FreeSurfer longitudinal stream[75]. Two participants were removed because their TBV estimate was larger than their ICV estimate. Final

distributions of TBV, ICV, and the LBA scores in 634 participants (49.8% females) are displayed in SFig. 3.

## MRI processing robustness checks in LBC1936

Details on MRI acquisition and processing software are listed per sample in Supplementary Data 1, indicating that different cohorts were processed with different FreeSurfer software versions. Although it was previously shown that different FreeSurfer software versions can affect study results (e.g., Gronenschild et al.[81]), we had planned this study expecting global measures of TBV and ICV to be within reasonable error margins as is typically assumed by large-scale imaging studies (e.g., Grasby et al.[82]). In the LBC1936 ($N = 581$), FreeSurfer v5 and FreeSurfer v7 created highly similar variance for TBV, ICV and LBA measures (Fig. S44, S45). It should be noted that we use the FreeSurfer eTIV measure to approximate ICV. eTIV is inferred through a linear transformation performed to align the image with a given atlas[83]. In general, studies consider eTIV a reasonable approximation of ICV. We show here that manual vs. FreeSurferv5 measures of ICV and TBV showed highly stable individual differences ($r = 0.89$; $r = 0.93$, respectively). Similar to a prior study[84], we find that eTIV is correlated with TBV above and beyond a manually estimated measure of ICV. Correlations between FreeSurferv5-based estimates and manually segmented estimates in the LBC1936 ($N = 631$) demonstrate that this bias is present with small effect size ($r_{\text{manually segmented ICV, FS-based LBA residual}} = 0.19$, $p = 1.56 \times 10^{-6}$; Fig. S46). LBA scores produced by the two estimation techniques yielded substantially correlated interindividual variability (LBA$_{\text{difference}}$: $r = 0.68$; LBA$_{\text{ratio}}$: $r = 0.63$, LBA$_{\text{residual}}$: $r = 0.70$; Fig. S47). Contrary to the FreeSurferv5-based CSF estimate, manually-segmented CSF in LBC1936 captured both the ventricle system and the subarachnoid space, which is in line with the substantial correlations between CSF$_{\text{manual}}$ and LBA ($r = |0.5-0.96|$; Fig. S46).

## Phenotype definitions

To characterise LBA and to compare it to longitudinally-observed atrophic changes, we calculated associations with health- and ageing-related traits. Those traits were derived based on the field IDs listed in Supplementary Data 2. Intercepts and slopes were modelled with growth curve models (growth function in lavaan). The code is displayed on GitHub (https://annafurtjes.github.io/BrainAtrophy_Genetics/).

## Computational approaches to estimate brain atrophy

We investigated different computational approaches to modelling brain atrophy either from one cross-sectional MRI scan or two repeated MRI scans acquired some years apart. Approaches using one scan leveraged ICV to approximate premorbid brain size. We refer to atrophy inferred from one cross-sectional MRI scans as LBA, and atrophy inferred from two longitudinal MRI scans as (within-person) *observed atrophic changes*. We report descriptive statistics for raw measures (Supplementary Data 6), but associations were calculated for standardised measures (mean = 0, SD = 1).

## LBA: Computing brain atrophy cross-sectionally using a single MRI scan

**Difference score.** The cross-sectional LBA difference score is computed on an individual-level from a cross-sectional MRI scan as the difference between ICV and TBV (*ICV-TBV*).

**Ratio score.** The cross-sectional LBA ratio (or proportional) score is computed on an individual-level from a cross-sectional MRI scan as the ratio between TBV and ICV (*TBV/ICV*).

**Residual score.** The cross-sectional LBA residual score is computed as the TBV-associated residuals from the regression between TBV and ICV (*TBV ~ ICV*). Raw residual scores are interpreted as the difference between an individuals' observed TBV and their predicted TBV given their ICV (i.e., a negative value means an individuals' TBV is smaller than expected given their ICV).

## Observed atrophic changes: Computing brain atrophy longitudinally using two repeated MRI scans

**Difference score.** The longitudinal difference score is computed on an individual-level from the difference between two estimates of TBV that were obtained from two time-shifted MRI scans (*TBV$_{\text{time 1}}$ − TBV$_{\text{time 2}}$*).

**Ratio score.** The longitudinal ratio (or proportional) score is computed on an individual-level from the ratio between two estimates of TBV that were obtained from two time-shifted MRI scans (*TBV$_{\text{time 2}}$/TBV$_{\text{time 1}}$*).

**Residual score.** The longitudinal residual score is computed as the TBV$_{\text{time 2}}$ associated residuals from the regression between TBV$_{\text{time 2}}$ and TBV$_{\text{time 1}}$ (*TBV$_{\text{time 2}}$ ~ TBV$_{\text{time 1}}$*).

## Statistical analyses

All *p*-values reported throughout were derived from two-sided statistical tests.

## Phenotypic analyses

**Phenotypic correlations and associations.** In the first phenotypic part of the paper, we report associations between LBA and health- and ageing-related traits (Fig. 2). Results were obtained with linear regressions when the outcome trait was continuous, with logistic regression when the outcome trait was binary, and with hurdle regression when the outcome trait was zero-inflated count data. Variance explained ($R^2$) in the outcome trait was calculated as the linear regression coefficient squared, $R^2$ was obtained with Nagelkerke's $R^2$ for logistic regression using the fmsb package in fmsb package, and a maximum likelihood pseudo $R^2$ for the hurdle regression using the pscl package in pscl package. *Beta* values are reported for predictor and outcome variables that were standardised to a mean of zero and a standard deviation of one. Phenotypic correlations were obtained via simple Pearson's correlations (cor.test function in R). All analyses were performed in R v4.2.2, and all plots were produced in ggplot2. Supplementary Data 3 summarises each phenotypic analysis, and links it with its respective methods and results section.

In the main analyses where LBA$_{\text{residual}}$ was assessed for associations with other variables (e.g., health-related outcomes or age), we used a two-step procedure where we derived the residuals first [*LBA$_{\text{residual}}$ = lm(TBV ~ ICV)*], and then tested the residuals for an associations with health-related outcomes or age [*lm(health trait ~ LBA$_{\text{residual}}$)*]. For comparison, we also tested those associations with a one-step procedure [*lm(other trait ~ TBV + ICV)*], which is what we mean when we refer to the 'adjustment method'. The unique contribution to $R^2$ by TBV in the adjustment method was calculated by subtracting the $R^2$ of a model excluding TBV from the full models' $R^2$.

## Genetic analyses

**Genetic data cleaning.** Of 45,598 UKB participants with available genetic and neuroimaging data, participants were excluded when labelled outliers in heterozygosity and missingness by the UKB analysis team (het_missing_outlier), and when they self-reported non-European ancestry or were missing this self-report information. European ancestry was also determined based on 4-means clustering of 40 genetic principal components. Relatedness was identified in ukbtools[85] using the default cut-off > 0.0884 King coefficient corresponding to 3rd degree relatedness. The sex check was performed in PLINK based on the observed number of heterozygote variants from that expected under Hardy-Weinberg equilibrium. Autosomal SNPs were filtered for missing genotype data at a rate of 0.02, minor allele frequency > 0.01

and Hardy-Weinberg equilibrium exact test at 0.00000001. Genotypes were imputed by the UKB analysis team with reference to the Haplotype Reference Consortium (HRC) and UK10K haplotype resource (Category 100319 (ox.ac.uk)). Numbers of included and excluded participants and SNPs are listed in Supplementary Data 4, 5.

**SNP-heritability.** We calculated SNP-based heritability ($h^2$) for TBV, ICV, CSF and all LBA measures using Genome-wide Complex Trait Analysis (GCTA v1.94.1; 30) in UKB genotype data with a cryptic relatedness cut-off at 0.025 ($N = 38,624$). The following covariates were included: age (field ID: 21022), sex (field ID: 31), assessment month (field ID: 53), assessment site (field ID: 54), x, y, and z scanner coordinates (field IDs: 25756, 25757, 25758), genotyping array, genotyping batch, and 40 genetic PCs. We did not include age-squared or product-wise age-by-sex as additional covariates because they were nearly perfectly correlated with age ($r = 0.99$) and sex ($r = 0.89$), respectively. Our reasoning was that including near identical variables would cause issues of multicollinearity which would produce unreliable estimates. Estimates of SNP-heritability were derived from a model with two variance components (common variants, and error), and estimates of SNP-by-age were derived from a model with three variance components (common variants, SNP-by-age interaction, and error).

**Genome-wide association study (GWAS) analysis.** GWAS were performed for TBV, ICV, CSF, and LBA inferred with the residual, ratio and difference method. Genome-wide SNP-based associations were calculated in a mixed linear model using REGENIEv3.4[32], including the following nuisance covariates: sex (field ID: 31), acquisition site (field ID: 54), acquisition month (field ID: 53), scan positions x,y,z (field IDs: 25756, 25757, 25758), 40 genetic PCs, genotyping array and genotyping batch. SNPs with minor allele count <5 and INFO score <0.4 were excluded. Given the wide age range in UKB, age (field ID: 21022) was also included as a covariate to ensure that atrophy levels were comparable between older and younger participants. Age-adjusted analyses should capture whether individuals show larger or smaller levels of atrophy in comparison with other individuals of similar ages and circumstances. We did not include age-squared or product-wise age-by-sex as additional covariates because they were nearly perfectly correlated with age ($r = 0.99$) and sex ($r = 0.89$), respectively. Our reasoning was that including near identical variables would cause issues of multicollinearity which would produce unreliable estimates. Additional GWAS were performed for males ($n = 20,453$) and females ($n = 22,657$) separately. Age-by-SNP interactions were also tested in REGENIEv3.4 using the --interaction flag.

The UKB resource provides 40 genetic PCs[86], which are commonly all included as covariates in neuroimaging GWAS using UKB data[46,47]. However, it was brought to our attention after we had pre-registered our analysis, that 40 genetic PCs likely overcorrect analyses, and thereby reduce power of GWAS analyses. Adjusting for >18 genetic PCs was shown to capture complex linkage disequilibrium in the full UKB sample[44]. We repeated analyses presented in Privé, Luu[44] in our neuroimaging subsample and identified that adjusting for 8 genetic PCs was the ideal number of genetic PCs to not overcorrect for patterns of linkage disequilibrium (*Supplementary Materials*). Hence, we re-calculated GWAS adjusting for 8 genetic PCs only, and compared the resulting GWAS summary statistics to those from the analysis with 40 genetic PCs. We also calculated another GWAS using the one-step adjustment method where TBV was the outcome phenotype and ICV was included in the list of covariates. All these different GWAS versions described here (i.e., difference, ratio, residual, adjustment method, 40 genetic PCs, and 8 genetic PCs) are made available on Zenodo, with exception of the SNP-by-age interaction results because they were null and did not provide information beyond the original GWAS results.

**Genetic correlations.** Using bivariate linkage disequilibrium score regression (LDSC;[45]), we used GWAS summary statistics calculated in the steps above to quantify genetic correlations with other structural neuroimaging and neurodegenerative traits. We only included HapMap3 SNPs (downloaded from the genomicSEM wiki; MHC region removed), and filtered for MAF > 0.01 and INFO > 0.9. SNP effects associated with the residual and ratio scores were flipped (i.e., multiplied by -1) so that a positive association can be interpreted as increasing the risk for LBA. We assessed the genetic correlations of these traits with other MRI-based phenotypes: TBV[46], TBV (BrainSegNotVentSurf, ID: 0167; 46), ICV[87], brain age[88], general dimensions of brain morphometry shared across 83 brain-wide volumes[60], brain ventricular volume[89]. Our pre-registration also included longitudinal changes in brain structure (Δ total brain;[8]), but this set of summary statistics produced a negative heritability estimate and was not suitable for LDSC. Finally, we used GWAS summary statistics of four neurodegenerative disorders associated with aging that were well powered enough to perform LDSC (i.e., $h^2$ Z-score > 4). The traits were Alzheimer's disease[49], Alzheimer's disease and related dementias[50], Parkinson's disease[52], amyotrophic lateral sclerosis[51]. We included the neurodegenerative traits post-hoc to provide a commentary on the current relevance of these genetic information to clinical outcomes.

**GWAS-by-subtraction model.** To obtain the TBV-associated residuals of ICV, we fitted a GWAS-by-subtraction model in GenomicSEM[54]. The lavaan syntax is printed in the Supplementary Materials (*Box S2*).

**Functional mapping and annotation (FUMA).** FUMA v.1.5.2 with default settings (32; https://fuma.ctglab.nl) was utilised to identify the genomic risk loci captured by our GWAS summary statistics (MAF < 0.01; INFO < 0.5). Independent significant SNPs were identified based on a genome-wide significance threshold ($p < 5 \times 10^{-8}$) and independence of other variants at $r^2 < 0.6$. LD structure was calculated from the UKB release2b 10k European reference panel population. From the set of independent significant SNPs, FUMA identified lead SNPs independent of one another at $r^2 < 0.1$. If identified SNPs were located close to another (< 250 kb), they were merged into one genomic risk locus, which means each locus can contain multiple independent significant SNPs and multiple lead SNPs. Gene prioritisation was performed based on both positional mapping (max. distance 10 kb), and expression quantitative trait loci (*eQTL*) mapping to identify whether independent significant SNPs from the GWAS (and their SNPs in LD) are known *eQTL*s in specific tissue types (refs. PsychENCODE, BRAINEAC, GTEx v8 Brain).

### Reproducibility statement

Analyses presented here were pre-registered at https://osf.io/gydmw/. Analysis code is presented at https://annafurtjes.github.io/BrainAtrophy_Genetics/.

### Reporting summary

Further information on research design is available in the Nature Portfolio Reporting Summary linked to this article.

## Data availability

The data analysed in this study are available for bona fide research purposes. We have obtained the appropriate data access permissions for each data source. Due to privacy and ethical considerations, the data cannot be deposited in a public repository. Researchers interested in accessing the data should directly contact responsible authorities managing each data source to obtain access permissions: UK Biobank (https://www.ukbiobank.ac.uk/), Lothian Birth Cohort 1936 (https://lothian-birth-cohorts.ed.ac.uk/data-access-collaboration), MRi-Share (https://datadryad.org/dataset/doi:10.5061/dryad.q573n5tj2), Human Connectome Project (https://www.humanconnectome.org/study/hcp-young-adult), Generation Scotland (https://genscot.ed.ac.uk/). GWAS

summary statistics produced by this study are available at https://doi.org/10.5281/zenodo.15282759.

## Code availability

The analysis only relied on open-source software and the code is displayed at https://annafurtjes.github.io/BrainAtrophy_Genetics/.

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

## Acknowledgements

We thank the participants who took part in LBC1936, UK Biobank, Human Connectome Project, MRi-Share and Generation Scotland, and the team members and radiographers who collected, entered, processed and disseminated the data used in this paper. LBC1936 MRI acquisition and initial analyses were conducted at the Brain Research Imaging Centre, Neuroimaging Sciences, University of Edinburgh (www.bric.ed.ac.uk), which is part of the SINAPSE (Scottish Imaging Network—A Platform for Scientific Excellence) collaboration (www.sinapse.ac.uk), funded by the Scottish Funding Council and the Chief Scientist Office. The LBC1936 is supported by the Biotechnology and Biological Sciences Research Council, and the Economic and Social Research Council [BB/W008793/1], Age UK (Disconnected Mind project), the Medical Research Council [G0701120, G1001245, MR/M013111/1, MR/R024065/1], the Milton Damerel Trust, and the University of Edinburgh. We acknowledge the work of the LBC1936 radiographers, particularly Gayle Barclay, Donna MacIntyre, Iona Hamilton, Charlotte Jardine who have enabled the high retention of the MRI component. The 1.5 T research MRI was funded by the Scottish Funding Council. We acknowledge the work of the LBC1936 radiographers Gayle Barclay, Donna MacIntyre, Iona Hamilton, Charlotte Jardine, who have enabled the high retention of the MRI component. We also thank Professor Stephen M. Smith for discussion on UK Biobank inter-site scanning differences. This research was conducted, using the UK Biobank Resource under approved project 10279. HCP data were provided by the Human Connectome Project, WU-Minn Consortium (Principal Investigators: David Van Essen and Kamil Ugurbil; 1U54MH091657) funded by the 16 NIH Institutes and Centers that support the NIH Blueprint for Neuroscience Research; and by the McDonnell Center for Systems Neuroscience at Washington University. Generation Scotland received core support from the Chief Scientist Office of the Scottish Government Health Directorates (CZD/16/6) and the Scottish Funding Council (HR03006) and is currently supported by the Wellcome Trust [216767/Z/19/Z]. This study was also supported and funded by the Wellcome Trust Strategic Award 'Stratifying Resilience and Depression Longitudinally' (STRADL) (Reference 525 104036/Z/14/Z). AF, IF & GD are supported by National Institutes of Health (NIH) grant [R01AG073593]. S.R.C. is supported by a Sir Henry Dale Fellowship jointly funded by the Wellcome Trust and the Royal Society (221890/Z/20/Z). EMTD was additionally supported by R01MH120219. EMTD is member of the Population Research Center (PRC) and the Center on Aging and Population Sciences (CAPS) at The University of Texas at Austin, which are supported by NIH grants P2CHD042849 and P30AG066614, respectively. W.D.H., and C.X. are supported by a Career Development Award from the Medical Research Council (MRC) [MR/T030852/1] for the project titled "From genetic sequence to phenotypic consequence: genetic and environmental links between cognitive ability, socioeconomic position, and health". J.W. is funded by the UK Dementia Research Institute which receives its funding from DRI Ltd, funded by the UK Medical Research Council, Alzheimer's Society and Alzheimer's Research UK. ADG is supported by NIH Grants R01MH120219 and R01AG073593. E.B. and K.F. are funded by the Stroke Association/BHF/Alzheimer's Society 'Rates Risks and Routes to Reduce Vascular Dementia' (R4VaD) Priority Programme Award in Vascular Dementia (16 VAD 07) and the Row Fogo Centre for Research into Ageing and the Brain (Ref No: AD.ROW4.35. BRO-D.FID3668413).

## Author contributions

Conception and design of the study: A.E.F., S.R.C., I.J.D. Data acquisition and curation (e.g., MRI processing or visual atrophy ratings): A.T., D.L., P.R., J.C., A.M.M., H.C.W., S.M.M., M.V.H., E.B., K.F., M.E.B., J.W., IJD, S.R.C. Data analysis and visualisation: A.E.F. Data interpretation: A.E.F., C.X., W.D.H., I.J.D., S.R.C. Coding: A.E.F. Provided GWAS summary stats for neurodegenerative diseases: I.F.F. Provided structural equation model code: J.M. Writing first draft: A.E.F. Writing and editing: A.E.F., I.F.F., GD, A.M.M. H.C.W., J.W., J.F., A.G., M.L., J.M., I.J.D., E.M.T., S.R.C.

## Competing interests

The authors declare no competing interests.
