## [Transparent Peer Review file · Nature Communications]

Measurement characteristics and genome-wide correlates of lifetime brain atrophy estimated from a single MRI

Corresponding Author: Dr Anna Furtjes

Version 0:

Reviewer comments:

Reviewer #1

(Remarks to the Author)

This manuscript examines the genetic correlates of lifetime brain atrophy, calculated using cross-sectional and longitudinal data (for which large sample sizes are typically infeasible). Several methods for estimating lifetime brain atrophy, based on the literature, are compared. The authors analyze correlations between atrophy and cognitive change, perform a genome-wide association study (GWAS) with lifetime brain atrophy as the outcome to estimate its heritability and identify genetic predictors.

The manuscript is well-written and provides a thorough exploration of this complex topic. The authors have clearly worked hard to produce comprehensive results, including the integration of data from multiple cohorts. Questions about how to best utilize cross-sectional neuroimaging data are particularly compelling, given the significant issues with selection bias in neuroimaging samples and representation in studies, which is compounded by loss to follow-up.

In the broader context of dementia research, this work highlights the importance of MRI measures, including atrophy, which are often underemphasized relative to amyloid and tau PET due to the focus on Alzheimer's pathology. These MRI measures are crucial in studying all-cause dementia, including vascular and non-Alzheimer's mechanisms. Attention to reproducibility in MRI-based studies is essential given this central role in studying all-cause dementia.

I have a few minor concerns that should be relatively easy to address:

Wang et al. and similar studies do not use the exact methods applied here. Specifically, the difference method employed does not align with the commonly used adjustment method, where intracranial volume is included as a covariate in a linear model. Additionally, Wang et al. did not calculate lifetime brain atrophy or use it as an outcome measure. Could you add adjustment models to align more closely with existing work? Furthermore, the terminology used in prior papers differs from "lifetime brain atrophy," even if they address similar constructs. It would be helpful to connect your work more precisely to the existing literature.

Wang (2024) notes issues with residualization, particularly regarding the selection of the referent group. Could you comment on this given that your residualized results align better with cognitive, functional, and atrophy-related outcomes? Were alternative reference groups used?

In the section on lifetime brain atrophy and aging-related health traits, please calculate confidence intervals for the differences between the residual, ratio, and difference methods if you wish to discuss their relative performance. If no significant differences are found, consider presenting results for atrophy calculated using other methods rather than focusing solely on the residual method. If making statements about differences across methods in Section 3, calculate and report differences with 95% confidence intervals. Given that methods are applied to the same dataset and are correlated, using a paired test, bootstrap approach, or the conservative z-test (as in Wang would) be appropriate.

In Figures 2c and 2d, it is unclear why APOE-e4 is treated as a binary trait when carriers of 1 and 2 alleles differ significantly. Multinomial logistic regression would be a better approach. For figure labeling, denoting e4 using the Unicode character ("u03B5") is what I find to be easiest R's ggplot.

The GWAS findings are intriguing and significantly enrich the paper. However, as I am not a geneticist, I cannot fully

evaluate the genetic methods.

As raised before—if differences between methods are discussed in the SNP heritability section, provide the differences and their 95% confidence intervals. Given the considerable overlap in the confidence intervals, it is unclear whether these differences are statistically meaningful. Again, the conservative z-test is probably the easiest option.

Overall, I thoroughly enjoyed this paper. Its comprehensive treatment of a challenging topic makes it a valuable contribution to the scientific literature.

(Remarks on code availability)

I reviewed the code for the non-genetic analyses focusing on the "Age-associated brain shrinkage" section. It's quite readable and seems to match their described analyses. The documentation is clear.

Reviewer #2

(Remarks to the Author)

The manuscript "Lifetime brain atrophy estimated from a single MRI: measurement characteristics and genome-wide correlates" by Fürtjes and colleagues investigates the genetic architecture of brain atrophy. In the same project, the authors try to assess the optimal way to determine brain atrophy from a single MRI scan, using intracranial volume as a marker of "what once was" and total brain volume as a marker of current brain volume. I was very excited to review this paper: in the genetics community people tend to think of (and combine) several measures like ICV, TBV or even head circumference at birth as being the same phenotype. The description and rationale in the introduction were an excellent read. It therefore pains me to say that in its present form, the paper is not fit for publication. The results are not easy to follow because of the many analyses - some performed but not considered relevant, with very little detail - and the method section feels somewhat incomplete. I think the paper would benefit greatly from using LBC1936 as a cohort in which to study the different measures of LBA, and UKB as a cohort in which to perform a GWAS analysis, and leave out all the other cohorts that were included but made interpretations more difficult out (see points 2,4 below). However, given that this was a preregistered study I am not sure whether simplifying the manuscript this way would be possible.

Major points:

1) While I do appreciate the mathematical description of the different brain atrophy concepts (e.g. Box 1), I do think that a biological discussion is lacking: there is a clear difference in interpretation between the difference score (brain atrophy as an absolute measure) and the other two (brain atrophy as a relative measure). This is likely also visible in the results from the GWASs, looking at the genetic correlations. There is a wealth of literature discussing this (e.g. in the context of cognitive reserve). I would have liked a thorough discussion of the underlying assumptions.

1b) Related to this point: "Single time-point MRI measures correlation with age" assumes a different association between LBA and age in different age ranges. That makes sense, and fits with a statement made on page 7: "Associations between longitudinally-observed atrophic changes and visually-rated atrophy were smaller [than with cross-sectional LBA] but isn't this testing something else (hence, comparisons between correlations are somewhat meaningless)? An association here would imply a linear relationship between the amount of atrophy at baseline and what happens longitudinally (i.e. some kind of acceleration process?) but contradicts assumptions made in other parts of the paper (e.g. Supplement, page 6 "assuming that brain atrophy declines continually and linearly.")

The interpretation of correlations between cross-sectional versus longitudinal measures of atrophy with another measure imply different shapes of age-trajectories and combining these measures in one plot makes it harder to interpret the data. Maybe the authors can consider to separate the two more explicitly (e.g. figures S3-S5).

1c) In general, I wonder what the rationale is for the analysis relating baseline volume to subsequent change. Supplement page 7: "Results do not seem intuitive and it is possible that estimates were unstable due to variance inflation, or other biases introduced by performing a regression that included both a derived measure (such as LBA difference, LBA ratio or LBA residual) and a baseline measure on which basis the first measure was derived (e.g., Butler et al., 2021; Glymour et al., 2005)." Doesn't the moderation analysis described in the previous paragraph suffer from this phenomenon as well?

Moreover, it seems like that analysis models an interaction without its main effect, it computes an interaction between baseline and subsequent change, which is questionable, and finally it uses the interaction effect as a predictor of one of its components (the dependent variable). Maybe I am mistaken in what analysis was done, but I strongly doubt any reliable conclusions can be drawn from this analysis. There probably is no perfect way to analyse baseline and subsequent change, but the approach used here seems flawed. In addition, I could not find this analysis in the code?

1d) Related to 1a): Page 17: "Furthermore, the difference method appears to have induced the counter-intuitive genetic correlation between greater LBA and larger TBV ($r_g = 0.37$; $SE = 0.04$)" and page 18: "We suggest a negative genetic correlation between LBA and TBV is theoretically more intuitive because a simple measure of TBV will likely capture, among other factors, both early-life neurodevelopment as well as later-life neurodegeneration (i.e., decline which is compatible with a negative direction of effects)" I am not sure whether I agree with this assessment, since the contribution of early life neurodevelopment is probably much larger than the neurodegeneration that follows in terms of the resulting TBV. This expectation seems to be related to the question whether brain atrophy can be expected to be an absolute number (in which case we would not expect a large association between the atrophy measure and the volume, or LBA to be a percentage loss - in which case we would expect larger brains to be associated with larger loss.

2) Similarly, a thorough discussion about ICV and how it can be measured is lacking. It looks like the authors have taken measures of ICV / TBV at face value. The descriptions of the different cohorts show that in each cohort ICV was measured with different software. Several cohorts use (different versions) of FreeSurfer, and moreover, use eTIV as a measure of ICV.

As noted by the creators of the Freesurfer processing suite, eITV is not a perfect estimate of ICV, see <https://freesurfer.net/fswiki/eITV> and its references, especially Klasson, N., Olsson, E., Eckerström, C. et al. Estimated intracranial volume from FreeSurfer is biased by total brain volume. *Eur Radiol Exp* 2, 24 (2018). <https://doi.org/10.1186/s41747-018-0055-4>. The latter manuscript, albeit in a relatively small sample, potentially changes the interpretation of the analyses and results. Given the above, I am not surprised that different cohorts seem to be incomparable. There is a similar issue with the definition of CSF: as far as I am aware, the CSF measure obtained from Freesurfer only captures the ventricle system. Outer CSF - cortical atrophy leading to bigger spaces between the gyri, is not included in this measure; linking LBA to CSF is therefore a very indirect comparison since the latter does not capture cortical atrophy. The authors seem to think it does: Supplementary Methods 2.1.1.1 "considering TBV volume equals ICV volume minus CSF volume." This strongly depends on the definition of CSF volume.

3) Results, LBA phenotype, Section 1: "and below we only report LBA results from the residual method because it consistently outperformed the difference and ratio method." Based on what is written here, I do not really see a difference between the different LBA measures. One of the strengths of this manuscript is the availability of an expert-defined atrophy measures, but the three LBA measures seem perform similarly. The results in Figure 2 do not really support the conclusion (the intervals of correlations between age-related phenotypes and all three LBA measure largely overlap).

4) Page 13: "extremely healthy nature of UKB participant's brains compared" this is a strong interpretation - not sure whether it is actually true. Wouldn't this be the result of the different measures used to assess ICV / TBV? (see point 2 above). Another potential explanation is that there seems to be no correction for sex in the phenotypic analyses (specifically relevant for the HCP dataset, page 11 "which seems to have been driven by 303 participants >31-years with small skulls that were excluded in the HCP data we analysed). Can this negative correlation be caused by a sex effect? Other papers on HCP show that males are overrepresented in the younger ages vs females overrepresented in the older group if I remember correctly.

5) Comparison of the GWAS results, page 16: "This suggests LBA ratio and LBA difference captured broader and less precise genetic signal compared to LBA residual." Why less precise? Since the measure are slightly different there is no way of knowing which loci should be found here. In addition: the genetic correlations are high and probably not different from 1? I would appreciate a little more information on the genetic loci and their overlap between the LBA measures. What genes do these loci map to? If the authors ran FUMA, it should be relatively easy to provide a bit more biological context to the GWAS hits.

6) Results, page 17: A genetical correlation of similar size (-0.37 versus 0.35) was described as weak and substantial, respectively. This feels like the authors are a bit biased in their interpretation.

Other points:

7) The results section is hard to read without referring to the methods (and even though a "methods" section is mentioned several times, I could not find it any of the documents - I received main manuscript, supplementary figures, supplementary methods and supplementary material (including supplementary results and supplementary information for the introduction). Specifically, there is no explanation of the intercepts and slopes in the health measures, apart from a brief mentioning in the STable 2.

8) Introduction: I think it would be good to mention that current genetic studies into brain size or ICV usually assume genetic factors on these phenotypes are the same - sometimes head circumference at birth is combined with adult TBV. I find the footnote on page 4 very relevant - I would not mind this seeing moved to the main text.

9) Page 7: "2) be greater in cross-sectionally scanned older adults (on average), and increase longitudinally with age"; What does increase longitudinally with age mean if not point 3)? Is it different from a linear effect of age on LBA?

10) Figure 2: Why was LBA computed at age 82? The authors claim to have no predefined notion of direction or causality - in the comment about APOE - , but I wonder why they try to predict previous decline (sCog, iCog - was the former defined at baseline, I would suppose so); using LBA at the oldest age.

11) Supplement aims 3.2 and 3.3: I am not sure whether creating the plots (and running a regression on) the variance explained is the way to go: by definition there will be more variance explained in the more elaborate model, and the p-value for the extra predictor will indicate whether this is a significant contribution. What is the value of correlation the two or regressing one onto the other? Do you expect exactly the same results for all health related phenotypes?

12) Discussion: why the use of absolute values? Does not help interpretation.

13) Supplementary Table 1: why are all the variances exact multiples of 1000?

14) Supplementary Figure S3: What is STRADL?- it is the Generation Scotland cohort I read from the methods. Please harmonise across the manuscript.

(Remarks on code availability)

I have looked at the code and it seems sufficient to assess what was done, I appreciate that the figures from the paper were

recreated there. Supplementary data / results do not seem to be included. I have not tried to rerun the analyses using the code.

Reviewer #3

(Remarks to the Author)

This paper addresses the interesting question of how much information about brain change can be extracted from a one-time point MRI. This was calculated from the relationship between TBV and ICV using different statistical methods and found to correlate modestly with neuroradiological atrophy rating scales ($r = .37-.44$) and longitudinally measured change ($r = .36$). Genetic analyses showed h^2 of 41%, with a strong polygenic signal.

There are many interesting aspects of the study. It addresses a well-known issue, but to my knowledge, it has not been shown with solid empirical data what the size of the correlation between longitudinally and cross-sectionally estimated atrophy actually is. The genetic analyses add to the story in a nice way. Personally, I find this much more convincing (and interesting) than the reported correlation with the neuroradiological rating scale results.

Also, the different relationships of LBA vs. 9-year longitudinal brain change to cognitive decline in the LBC1936 sample are highly interesting. I have never seen anything like that reported previously, directly contrasting current changes vs. estimated accumulated lifetime changes. This is, in my opinion, a very strong aspect of the study!

The different ways of correcting for ICV have been thoroughly investigated before, and the residual approach has been widely regarded as the most optimal one, which aligns with the present results. The LBA measure captures approximately 13% of the variance in actually measured brain change as measured by longitudinal MRIs over 9 years. This is relatively modest but still shows a substantial degree of overlap. The maximal correlation with longitudinal estimated atrophy will (of course) ultimately depend on the reliability of the measures. Especially for longitudinal estimates, this is a real limitation, particularly for relatively short follow-up intervals as used in, e.g., UKB (see, e.g., <https://www.biorxiv.org/content/10.1101/2024.06.03.592804v1>). I think this is a relevant issue for the current paper; maybe it could shortly be discussed or mentioned? The nine-year follow-up interval in the LBC1936 is a real strength in this regard. I think the discussion in general is well-balanced and reflects the nature of the results well, but maybe some more attention could be devoted to the strength of the relationship between the measure of "lifetime atrophy" (LBA) vs. the measure of "current atrophy" (longitudinal change)?

A limitation of this approach, of course, is that only global atrophy can be assessed, but this is discussed as a limitation.

I think the statistics section could include some more information to make the reading easier, as it is presented before the methods. For example, in the analyses of the relationship between LBA and cognitive measures, were any covariates included? Since sex is strongly associated with ICV and TBV, it would be nice to see some results showing that sex did not associate with the different LBA measures (probably does for the difference measure, but maybe not for the residual measure, for example).

I was not sure why LBA relations to brain age gaps were reported, as this is another cross-sectional measure, which has been shown to be very weakly related to actual brain change.

A small comment: It is mentioned in the introduction that a problem with longitudinal MRIs is that it is inconvenient and hence limits statistical power, e.g., for genetic studies. I would suggest adding that in a lifespan context, it is currently almost impossible to cover more than a couple of decades using longitudinal MRIs, and a "one-time point" atrophy estimation would then be the only realistically attainable option.

The reported change in TBV over 9 years in LBC1936 of approximately 0.75% annually seems perfectly reasonable for this age group, aligning with previous reports of cognitively healthy older adults.

Minor 1: Is the "age-moderated brain shrinkage" mentioned in the abstract the associations with the brain age gap, chronological age, or something else? It would be good to consider another formulation, I believe, as it was not clear to me what this refers to.

Minor 2: It is stated regarding the genetic analyses that "we performed genome-wide investigations in over 2.5 times more participants." This may strictly speaking be true, but as discussed in the manuscript, other studies have controlled for ICV in the GWAS analyses, which would yield results with some similarities to the residualized approach used here. On the other hand, given the correlation of .36 with actual change, the LBA measure cannot be equated with longitudinal brain change as measured by repeated MRIs. Hence, I am not sure the "2.5-statement" is very relevant.

Minor 3: Is it accurate to describe $h^2=41%$ as "strongly heritable"?

(Remarks on code availability)

Version 1:

Reviewer comments:

Reviewer #1

(Remarks to the Author)

I would still like to see the adjustment method included. Wang concluded this was probably the best method in that it was easiest to implement. Wang also raised significant issues with the residual method. Given arguments in favor of the adjustment method over other methods, including the residual method, it seems like a major omission to exclude it.

I am not sure I understand why producing two coefficients for the adjustment method is an issue. Regression coefficients are proportional to correlation coefficients and produce the same standard errors. So coefficients can be compared in lieu of correlations without issue.

If statistical power is a limitation, a bootstrap could be used in lieu of a conservative z test. I'd maintain that if one want to discuss differences between methods, one needs to calculate those differences and a corresponding uncertainty measure.

(Remarks on code availability)

Reviewer #2

(Remarks to the Author)

The revision of "Lifetime brain atrophy estimated from a single MRI: measurement characteristics and genome-wide correlates" by Fürtjes and colleagues reads much better than the first version, and the authors really made an effort to answer my questions and add clarification where needed.

I only have a few minor points left:

"Line 441: "It is notable that the residual score was constructed entirely independent of ICV" I am not sure what this means given that LBA_resid equals the residual after regressing out ICV?"

I would suggest the authors take another look at the consistency of the direction of effects. E.g page 18 states ". The same measure of LBA residual was strongly associated with the TBV GWAS by Smith et al. (2021) (r_g residual = -0.81; SE = 0.06; negative r_g because direction of effects flipped for LBA residual)", while at the beginning they also report a moderate genetic correlation with TBV "LBA residual had a moderate negative genetic correlation with our TBV GWAS (r_g residual = -0.37; SE = 0.05)" This one does not seem to be flipped - or it is not stated that this was done, but the direction is the same? The statement " r_g across three LBA scores = 0.72-0.96" later on page 18 suggests that the sign was flipped for these analyses. One solution to all of this would be to mention once that all LBA measures were flipped to indicate "higher values equal more brain atrophy", adjust section 2.2.1 accordingly and adjust signs throughout the paper. This way the absolute values in the discussion are not needed - to be honest, using them still raises red flags for me, as if the signs do not match expectations, which I believe they do - and all other mentions of flipping that make the reader stop and question why could be removed, which would improve readability.

(Remarks on code availability)

I have not reviewed the latest version of the code in detail, it looks complete and well documented.

Reviewer #3

(Remarks to the Author)

The authors have responded to my comments in an excellent way, and the manuscript is in my opinion in ready for publication.

(Remarks on code availability)

Version 2:

Reviewer comments:

Reviewer #1

(Remarks to the Author)

The authors have thoroughly addressed all of my comments, and I appreciate the inclusion of the additional results. This is a strong and important manuscript.

(Remarks on code availability)

Reviewer #2

(Remarks to the Author)

No further comments.

(Remarks on code availability)

I reviewed the code in a previous submission.

REVIEWER COMMENTS

We would like to thank the editor and the reviewers for the opportunity to revise our manuscript. The comments and feedback have been very insightful, and we believe that the changes have considerably improved the manuscripts rigour and clarity. A point-by-point rebuttal is provided below, where our responses are printed in red. The indicated line numbers refer to the manuscript version with tracked changes.

In addition to the reviewer's suggestions below, we have added one supplementary analysis. It has been brought to our attention by a colleague that adjusting GWAS for 40 genetic PCs may reduce power because some genetic PCs were shown to capture complex linkage disequilibrium rather than complex population structure (the latter being the reason we adjust for PCs). Using the tool provided by Privé et al. (2020), we now present empirical evidence that the UKB neuroimaging subset used in our study is likely best adjusted for 8 instead of 40 genetic PCs, and present this evidence in the Supplementary Materials (under *GWAS re-analysis adjusting for 8 genetic PCs only*). As a sensitivity analysis, we have repeated the GWAS analyses from the main analysis, adjusting for 8 instead of 40 genetic PCs, and discuss in line 403 the slight impact this had on the presented results:

“Non-pre-registered investigations using the tool by Privé et al. (2020) delivered empirical evidence that only 8 genetic PCs should be included for this UKB neuroimaging subset (*SMaterials; GWAS re-analysis adjusting for 8 genetic PCs only*). Adjusting for 8 instead of 40 genetic PCs did not affect LBA_{residual}, but LBA_{difference} and LBA_{ratio} captured more GWAS loci (*Table S9*), likely reflecting increased power by only including PCs that showed no evidence for capturing complex LD structure.”

We will make publicly available GWAS summary statistics for both GWAS adjusted for 40 as well as 8 genetic PCs.

Reviewer #1 (Remarks to the Author):

This manuscript examines the genetic correlates of lifetime brain atrophy, calculated using cross-sectional and longitudinal data (for which large sample sizes are typically infeasible). Several methods for estimating lifetime brain atrophy, based on the literature, are compared. The authors analyze correlations between atrophy and cognitive change, perform a genome-wide association study (GWAS) with lifetime brain atrophy as the outcome to estimate its heritability and identify genetic predictors.

The manuscript is well-written and provides a thorough exploration of this complex topic. The authors have clearly worked hard to produce comprehensive results, including the integration of data from multiple cohorts. Questions about how to best utilize cross-sectional neuroimaging data are particularly compelling, given the significant issues with selection bias in neuroimaging samples and representation in studies, which is compounded by loss to follow-up.

In the broader context of dementia research, this work highlights the importance of MRI measures, including atrophy, which are often underemphasized relative to amyloid and tau PET due to the focus on Alzheimer's pathology. These MRI measures are crucial in studying all-cause dementia, including vascular and non-Alzheimer's mechanisms. Attention to reproducibility in MRI-based studies is essential given this central role in studying all-cause dementia.

I have a few minor concerns that should be relatively easy to address:

Wang et al. and similar studies do not use the exact methods applied here. Specifically, the difference method employed does not align with the commonly used adjustment method, where intracranial volume is included as a covariate in a linear model. Additionally, Wang et al. did not calculate lifetime brain atrophy or use it as an outcome measure. Could you add adjustment models to align more closely with existing work?

Thank you for bringing this lack of clarity to our attention. Overall, we had identified correction methods for measuring atrophy from various sources specific to that endeavour (e.g., *Box S1*). Nevertheless, the approaches used specifically in Wang et al. (whose main motivation was to adjust for ICV as a confound variable rather than model atrophy) largely overlap with ours.

We had included the Wang et al. reference as an example of a recent, large-scale paper that used both the ratio ('proportional') and a regression-based method. The author is quite right: Wang et al. do not use the difference method (because they are not modelling change), and we do not use the 'adjustment' method (i.e., ICV added as a covariate in the model testing association between TBV and another trait, instead of extracting residuals first and then testing the association with another trait). We agree it would be helpful to clarify this, and have done so in the introduction (line 102):

"All three computational approaches have previously been used (e.g., Brower et al., 2022 using difference method; Wang et al., 2024 using ratio and residual methods to adjust for ICV),"

Given our study aims to first characterise and second compare external associations for each LBA method, unfortunately the 'adjustment' method used by Wang et al. does not fulfil our requirements to support a direct comparison between the three LBA scores. This is because $\text{lm}(\text{trait} \sim \text{residual score})$ produces one regression coefficient, while $\text{lm}(\text{trait} \sim \text{TBV} + \text{ICV})$ produces two regression coefficients, prohibiting a direct comparison with the difference and the ratio methods. As such, we judge that the three LBA phenotypes we have already included could be compared most straightforwardly by modelling standalone scores first, then describing their measurement characteristics, and then performing a GWAS using the same covariates across all three LBA phenotypes. While we appreciate that Wang et al. show that the residual and the adjustment method can produce varying trait predictions, we feel that these two approaches should at least theoretically be identical (for example argued in Dalecki & Willits, 1991).

Furthermore, the terminology used in prior papers differs from "lifetime brain atrophy," even if they address similar constructs. It would be helpful to connect your work more precisely to the existing literature.

We agree it is important to refer to other terms that the literature has previously used for estimates obtained from the ratio method (e.g., brain parenchymal fraction), and have added a sentence to the introduction to refer the reader to the terms in *Box S1*:

Line 99: "Other studies refer to the *brain parenchymal fraction* when using the ratio method (*Box S1*, Rudick et al., 1999)."

We hope that the statement below (discussion) will help the reader to distinguish between our study (modelling atrophy) and other studies aiming to simply remove the confounding effect of ICV which is not necessarily geared towards modelling atrophy.

Line 498: "Our study explicitly demonstrates that the TBV-associated residuals adjusted for ICV capture ageing-related variance (in older cohorts) that TBV or ICV variables could not

have captured in isolation. This goes beyond simply adjusting for (potentially) confounding effects, as is often given as justification for ICV adjustment (Wang et al., 2024).’

Wang (2024) notes issues with residualization, particularly regarding the selection of the referent group. Could you comment on this given that your residualized results align better with cognitive, functional, and atrophy-related outcomes? Were alternative reference groups used?

As outlined in *Box S1*, we agree with the reviewer that using an appropriate reference group is one important feature of using the residual method. This issue is relevant for small samples due to overfitting, or because the regression line could be influenced by extreme outliers, or where other sample characteristics (such as age, as in Wang et al) differ. To deal with this, we have used reasonably large cohorts, inspected the TBV and ICV regression for extreme outliers (*SFig.1*), and paid specific attention to the age range of the reference group in these analyses.

With regard to reference groups, it had been our main priority to maximise sample size in order to obtain the most robust regression line. It is an important point that using a different reference group (other than the full sample) could impact the residual-based estimates, especially when different age groups are used as shown in Wang et al. In response to a comment by another reviewer, analyses presented in Section 1-3 have been repeated in males and females separately, including the derivation of the residual score with reference to males or females only. This produced largely consistent results to those in the main analyses, and indicates that using males and females as separate reference groups did not impact variance captured by the residual score. We have added a line to the limitation section that future studies should assess the impact of using reference groups of different ages (line 581):

“Our residual scores were constructed with reference to full samples (or males and females in sex-split analyses) where selecting appropriate age ranges was shown to influence results (Wang et al. 2024). Future work is needed to assess the impact of using different reference samples.”

The reviewer also asked us to comment on the divergent UKB associations with cognitive tests reported in Wang et al. Wang et al. demonstrate (in their *Fig. 4*) that the association between residualised TBV and individual cognitive traits increases when residual scores were derived based on successively older reference samples. We would argue that this is in keeping with our results displayed in *Fig.3* where we display that LBA is moderated by sample age, and that the correlation between LBA and age gets stronger, the older the sample. The older the subsample, the more brain shrinkage (i.e., larger LBA). Simultaneously, cognitive functions decline with older age, which is likely due to ageing-related processes that we hope to capture with LBA. With regard to Wang et al., we propose that such intercorrelations (e.g., between age, cognitive function, and brain shrinkage) likely drive the very slight increases in regression coefficients with older reference groups (displayed in *Fig.4* in Wang et al.), which may reflect that a larger proportion of variance will be attributable to ageing and neurodegenerative differences as a function of increasing age.

Finally, we can confirm that we find slightly larger associations (between cognitive ability and TBV adjusted for ICV) than Wang et al. (our *Fig.S7*), even though both analyses are using the UKB sample. Our intuition is that those differences stem from different treatments of the UKB cognitive variables between Wang et al.’s and our study: Wang et al. modelled associations of individual cognitive tests while we modelled a *g*-factor across the individual tests. The *g*-factor allowed us to extract ‘error-free’ variance shared across all those individual tests which likely amplified the signal relevant to brain shrinkage. We hope that the modified statement in line 226 makes this more explicit:

“In the UKB sample, LBA was significantly associated with greater general cognitive function (***g*-factor**; $\beta_{\text{residual}} = 0.23$ [95%CI = 0.20-0.26]) [...]

In the section on lifetime brain atrophy and aging-related health traits, please calculate confidence intervals for the differences between the residual, ratio, and difference methods if you wish to discuss their relative performance. If no significant differences are found, consider presenting results for atrophy calculated using other methods rather than focusing solely on the residual method. If making statements about differences across methods in Section 3, calculate and report differences with 95% confidence intervals. Given that methods are applied to the same dataset and are correlated, using a paired test, bootstrap approach, or the conservative z-test (as in Wang would) be appropriate.

Thank you for this comment. In Section 3, our intention was not to formally compare the correlation magnitudes and we refrain from making any statement about their differences (line 309): Instead of ‘For all computational methods, the Pearson’s correlation was modest, but numerically highest when atrophy was inferred with the residual method’ we now state ‘For all computational methods, the Pearson’s correlation was modest’.

In Section 2, we agree that it is important to test whether statistical significance supports our claim that the residual score, on average, predicts other ageing-related traits better than the difference and ratio scores. Given our hypotheses are based on the aggregation of association magnitudes across all analyses in this section, we provide supplementary analyses appropriate to make conclusions about overall numerical trends that we are concerned with in Fig.2. We plot R^2 estimates from the residual method against those from the ratio and difference method (Fig.S11; Fig.S13), to fit a linear regression that describes the relationship between R^2 obtained from the residual method vs. R^2 from each of the other methods. Including all 15 outcome traits analysed in Section 1, the 95% confidence intervals indicate that the slope is not statistically significantly different from one, meaning that $LBA_{residual}$ does not produce overall numerically larger R^2 estimates than $LBA_{difference}$ or LBA_{ratio} . Cook’s distance analyses illustrated that the cross-sectionally-estimated outcome measures like brain age and visually rated atrophy were outliers in this analysis. The 95% confidence intervals were significantly different from 1 for the comparison between $LBA_{residual}$ and $LBA_{difference}$ (but not LBA_{ratio}) when excluding the outlier traits, illustrating that outlier traits skewed the trend to be non-significant when all outcome traits were included.

Line 205: “There was a significant aggregated trend to support that $LBA_{residual}$ explained more variance (R^2) across all estimates than $LBA_{difference}$ ($p = 0.0001$; not LBA_{ratio} $p = 0.085$), but only when excluding outlier traits (Cook’s distance $> 4/15$). Outlier traits were visually rated atrophy (deep and superficial) and brain age; the three cross-sectionally-derived neuroimaging traits included in this analysis (Fig.S11; Fig.S13 for UKB).”

We would like to add that, with this sample size ($N = 286$), power would not have been sufficient for pairwise comparisons: If we had performed pairwise comparisons of effect sizes from the residual score against effect sizes from the other two methods (30 significance tests), each difference in effect size would have to be larger than Cohen’s $d = 0.24$ for our analysis to achieve 80% at $p = 0.05/30$ (two-sided z-test). We can see numerically in Fig.2 that the differences are much smaller and that it is not realistic to detect any significant differences with our setup.

Fig.S11. LBC1936: Linear relationships between R^2 estimates displayed in Fig.2 to contrast whether LBA residual predicted outcome traits consistently better than LBA difference or LBA ratio (left). Significance levels to determine whether the linear slope was significantly different from one were: $p = 0.454$ for LBA difference and $p = 0.723$ LBA ratio including outlier traits. Cook's distance (right) indicates outlier traits that negatively impacted the fit of the regression line. Significance levels to determine whether the linear slope was significantly different from one for regression excluding outlier traits were: $p = 0.0001$ for LBA difference and $p = 0.085$ LBA ratio. Values in square brackets indicate 95% confidence intervals.

In Figures 2c and 2d, it is unclear why APOE-e4 is treated as a binary trait when carriers of 1 and 2 alleles differ significantly. Multinomial logistic regression would be a better approach.

This is a very helpful comment, thank you. We agree that modelling the number of APOE $\epsilon 4$ alleles as a multilevel variable would be a better approach. Unfortunately, the LBC1936 participants that were assessed at wave 5 include only five participants with two APOE $\epsilon 4$ alleles, which we suggest are too few participants to ensure reasonable model precision and stability. Instead, we have added a visual representation of LBA distributions by number of APOE $\epsilon 4$ alleles in the supplement (SFig.10).

Fig.S10. LBC1936: LBA distributions displayed by number of APOEε4 alleles.

The same plot for UKB participants is in *Fig.S12*. Contrary to the LBC1936, we have enough UKB participants in all APOE categories to perform a multinomial regression (0 alleles = 2499; 1 allele = 877; 2 alleles = 69). The results from this multinomial regression indicated no significant associations between APOEε4 (3 categories) and LBA as a predictor variable. That is, the coefficients for having 1 & 2 APOEε4 alleles (representing the difference in log-odds of having 1 or 2 APOEε4 alleles relative to having 0 APOEε4 alleles) were not significant:

$APOE\epsilon4 \sim LBA_{residual}$: coefficient for having 1 allele = -0.04 ($SE = 0.13$; $p = 0.737$); coefficient for having 2 alleles = -0.06 ($SE = 0.13$; $p = 0.622$)

$APOE\epsilon4 \sim LBA_{ratio}$: coefficient for having 1 allele = -0.09 ($SE = 0.13$; $p = 0.465$); coefficient for having 2 alleles = -0.10 ($SE = 0.13$; $p = 0.442$)

$APOE\epsilon4 \sim LBA_{difference}$: coefficient for having 1 allele = 0.18 ($SE = 0.13$; $p = 0.162$); coefficient for having 2 alleles = 0.17 ($SE = 0.13$; $p = 0.203$)

These null results fit in with the null associations we had obtained as part of our GWAS with each of the two individual *APOE* SNPs rs7412 and rs429358 (line 389).

Fig.S12. UKB: LBA distributions displayed by number of APOEε4 alleles. Multinomial regressions treating APOEε4 as a categorical outcome variable yielded non-significant associations with LBA scores. Coefficients indicate the change in log-odds for having 1 or 2 APOEε4 relative to having 0 APOEε4 alleles. APOEε4 ~ LBA_{residual}: coefficient for having 1 allele relative to 0 alleles = -0.04 (*SE* = 0.13; *p* = 0.737); coefficient for having 2 alleles = -0.06 (*SE* = 0.13; *p* = 0.622). APOEε4 ~ LBA_{ratio}: coefficient for having 1 allele = -0.09 (*SE* = 0.13; *p* = 0.465); coefficient for having 2 alleles = -0.10 (*SE* = 0.13; *p* = 0.442). APOEε4 ~ LBA_{difference}: coefficient for having 1 allele = 0.18 (*SE* = 0.13; *p* = 0.162); coefficient for having 2 alleles = 0.17 (*SE* = 0.13; *p* = 0.203).

For figure labeling, denoting e4 using the Unicode character ("u03B5") is what I find to be easiest R's ggplot.

Thank you for being so specific about this. We have added the Unicode character to *Fig.2*. We agree it looks better.

The GWAS findings are intriguing and significantly enrich the paper. However, as I am not a geneticist, I cannot fully evaluate the genetic methods.

As raised before—if differences between methods are discussed in the SNP heritability section, provide the differences and their 95% confidence intervals. Given the considerable overlap in the confidence intervals, it is unclear whether these differences are statistically meaningful. Again, the conservative z-test is probably the easiest option.

We have now added a conservative z-test to the SNP-heritability section (line 353):

“The SNP-heritability of the residual score was significantly lower (41% [95% CI = 38-43%]) than for either the ratio (42% [95% CI = 40-44%]) or difference score (47% [95% CI = 45-49%]), when testing their differences with a conservative z-test accounting for their genetic correlations (LBA_{difference}: $z = 3.54$, $p = 0.0002$; LBA_{residual}: $z = 8.02$, $p = 5.38 \times 10^{-16}$).”

Overall, I thoroughly enjoyed this paper. Its comprehensive treatment of a challenging topic makes it a valuable contribution to the scientific literature.

Reviewer #1 (Remarks on code availability):

I reviewed the code for the non-genetic analyses focusing on the "Age-associated brain shrinkage" section. It's quite readable and seems to match their described analyses. The documentation is clear.

We greatly appreciate your thorough engagement with our study, thank you.

Reviewer #2 (Remarks to the Author):

The manuscript "Lifetime brain atrophy estimated from a single MRI: measurement characteristics and genome-wide correlates" by Fürtjes and colleagues investigates the genetic architecture of brain atrophy. In the same project, the authors try to assess the optimal way to determine brain atrophy from a single MRI scan, using intracranial volume as a marker of "what once was" and total brain volume as a marker of current brain volume. I was very excited to review this paper: in the genetics community people tend to think of (and combine) several measures like ICV, TBV or even head circumference at birth as being the same phenotype. The description and rationale in the introduction were an excellent read. It therefore pains me to say that in its present form, the paper is not fit for publication.

The results are not easy to follow because of the many analyses - some performed but not considered relevant, with very little detail - and the method section feels somewhat incomplete. I think the paper would benefit greatly from using LBC1936 as a cohort in which to study the different measures of LBA, and UKB as a cohort in which to perform a GWAS analysis, and leave out all the other cohorts that were included but made interpretations more difficult out (see points 2,4 below). However, given that this was a preregistered study I am not sure whether simplifying the manuscript this way would be possible.

This is a helpful comment, thank you. We have reworded the beginning of our Results section to give more methodological detail before we describe the results (lines 138 onwards). We hope this is now easier to follow. For a better overview of the analyses performed, we have added an additional supplementary table (*Table S3*) that outlines all analyses referenced in the paper, in order to link together results and methods. The table explicitly states analysis purpose and interpretation, R functions, and cohorts used to perform each analysis. Hopefully, this will make the methods section feel more complete. We would like to keep the results from all included cohorts, because we value honest and transparent reporting of results including all pre-registered analyses even when that means the manuscript is somewhat harder to digest.

Major points:

1) While I do appreciate the mathematical description of the different brain atrophy concepts (e.g. Box 1), I do think that a biological discussion is lacking: there is a clear difference in interpretation between the difference score (brain atrophy as an absolute measure) and the other two (brain atrophy as a relative measure). This is likely also visible in the results from the GWASs, looking at the genetic correlations. There is a wealth of literature discussing this (e.g. in the context of cognitive reserve). I would have liked a thorough discussion of the underlying assumptions.

We agree that a biological discussion of the LBA phenotypes is important. We have now updated the discussion to cover the conceptual / biological differences between the LBA measures, and more clearly link this to brain and cognitive reserve in the context of the phenotypic and genetic associations. The Discussion section entitled 'Significance of carefully considered methodological approaches' has been updated to read as follows:

"The work presented here highlights the importance of carefully considering and interpreting different methodological approaches used to estimate lifetime changes (i.e., difference, ratio, residual method) which includes adjusting for the appropriate covariates in the GWAS analysis. Conceptually, the different LBA estimates make different underlying assumptions that influence their biological interpretation. The difference score reflects an absolute measure of total volume loss from an inferred baseline, which does not account for individual differences in initial adult size which may be influenced by sex and neurodevelopmental factors. This contrasts with relative measures (though the reference group remains an

important consideration). As we show, only atrophy calculated with the residual method will be genetically and phenotypically uncorrelated with ICV baseline differences, and atrophy calculated using the difference or ratio method highly likely comprises results driven by ICV baseline differences. Hence, the difference and ratio methods tag genetic signal that amalgamates late-life neurodegeneration (or any longitudinal changes), as well as developmental processes indicated by ICV (r_g between ratio score and ICV = |0.35|; r_g between difference score and ICV = |0.75|). Moreover, the phenotypic and genetic results are in keeping with the literature on brain and cognitive reserve (Stern et al., 2020). Though all three LBA measures reflect some aspect of brain reserve (the raw quantity of cerebral resource that allows a given brain to better tolerate neuropathology), their correlations with ICV indicate an important distinction between ‘static’ and ‘dynamic’ reserve (van Loenhoud et al., 2018). LBA measures that are more strongly correlated with ICV (e.g., for LBA_{difference}) are more reflective of pre-existing brain and head size differences from when they were maximal (‘static reserve’). By contrast, zero-correlations with ICV (e.g., for LBA_{residual}) can be considered a marker of an individual’s dynamic brain reserve at the time of the scan.”

1b) Related to this point: “Single time-point MRI measures correlation with age” assumes a different association between LBA and age in different age ranges. That makes sense, and fits with a statement made on page 7: "Associations between longitudinally-observed atrophic changes and visually-rated atrophy were smaller [than with cross-sectional LBA] but isn't this testing something else (hence, comparisons between correlations are somewhat meaningless)? An association here would imply a linear relationship between the amount of atrophy at baseline and what happens longitudinally (i.e. some kind of acceleration process?) but contradicts assumptions made in other parts of the paper (e.g. Supplement, page 6 "assuming that brain atrophy declines continually and linearly.")

The interpretation of correlations between cross-sectional versus longitudinal measures of atrophy with another measure imply different shapes of age-trajectories and combining these measures in one plot makes it harder to interpret the data. Maybe the authors can consider to separate the two more explicitly (e.g. figures S3-S5).

We apologise that this has not been clear in the manuscript. The reviewer is referring to multiple distinct analyses here. It is correct that we perform analyses to understand the association between LBA and age (Section 2; first available neuroimaging visit to maximise sample size), which is distinct from the analyses performed to understand the association between LBA and longitudinally observed atrophic changes (Section 3; LBA at final neuroimaging visit and longitudinal estimate extracted from first and final neuroimaging visit). Also Section 1 is distinct: analyses consider LBA estimates at the latest available time point (i.e., wave 5 for LBC, wave 3 for UKB), and the visually-rated atrophy ratings from the latest available time point (wave 3). This analysis does *not* consider atrophy at baseline.

Line 257: “Visually-rated atrophy was assessed at age 76 (wave 3) which was the last available time point. Ideally, we would have preferred to include visually-rated atrophy at age 82 (wave 5), so that LBA, the final time point of longitudinal atrophic changes, and visually-rated atrophy would have been recorded at the same time point.”

We have modified the introductory paragraph to our results section in order to make it clearer that each section answers a distinct hypothesis. *Table S3* should help to make it more obvious that they are separate. At no point did we intend to compare the correlations across those three sections, because they answer distinct hypotheses utilising measures at different time points. In line with the reviewer’s comment, we do not imply any process of acceleration which is reflected in our choice of linear statistical tests. We have made this more explicit in the limitations (line 584):

“Our use of linear statistical methods implies linear processes, which we argue is a reasonable assumption for the short time windows each cohort covers, but this may not be sufficient for lifespan studies.”

We verified that the findings from each of the separate hypotheses are only interpreted as complimentary characteristics of LBA, and that they are not compared in their magnitudes (e.g., line 464).

1c) In general, I wonder what the rationale is for the analysis relating baseline volume to subsequent change.

The analysis under ‘Associations with ageing-related variables’ (Supplementary Materials; lines 42 onwards) was pre-registered to compare predictive performances across different nested models. For example, Aim 3.2 was designed to test whether LBA predicts trait variance above and beyond variance explained by TBV alone (or whether their explained variance is overlapping). In order to test this, we fitted two nested models: first, an association between an outcome trait and TBV + LBA, and second, an association between the same outcome trait and TBV alone. A significant result from a χ^2 comparison of the nested models indicates that adding LBA to the model significantly improved the model fit given the loss in degrees of freedom, under the null hypothesis that the simple model is better. This would suggest that LBA captures trait-relevant variance that TBV alone could not capture. We would have taken this as evidence that LBA adds meaningful information above and beyond only considering TBV (see pre-registration: <https://osf.io/gydmw/>).

Supplement page 7: “Results do not seem intuitive and it is possible that estimates were instable due to variance inflation, or other biases introduced by performing a regression that included both a derived measure (such as LBA difference, LBA ratio or LBA residual) and a baseline measure on which basis the first measure was derived (e.g., Butler et al., 2021; Glymour et al., 2005).” Doesn’t the moderation analysis described in the previous paragraph suffer from this phenomenon as well? Moreover, it seems like that analysis models an interaction without its main effect, it computes an interaction between baseline and subsequent change, which is questionable, and finally it uses the interaction effect as a predictor of one of its components (the dependent variable). Maybe I am mistaken in what analysis was done, but I strongly doubt any reliable conclusions can be drawn from this analysis. There probably is no perfect way to analyse baseline and subsequent change, but the approach used here seems flawed. In addition, I could not find this analysis in the code?

We appreciate the thorough engagement with our Supplementary Materials. We agree with the reviewer that this is just a fundamentally difficult thing to model well, and this issue certainly does extend to the moderation analyses. We think the reviewer is right in suggesting that including the latter causes more confusion than we gain from being transparent with respect to the pre-registration. In addition to the multicollinearity issues, the limited variance captured by longitudinally-estimated atrophy prohibited the moderation analysis to produce interpretable results, and we have therefore removed the moderation results from the Supplementary Materials. Under ‘Deviations from the pre-registration’ (*SMethods*) we give these reasons to explain why we removed this analysis from the Supplementary Materials.

Upon the reviewer’s request, we have now included the code for supplementary analyses (aims 3.1-3.3) to be more transparent, even about the analyses we chose not to interpret.

1d) Related to 1a): Page 17: “Furthermore, the difference method appears to have induced the counter-intuitive genetic correlation between greater LBA and larger TBV ($rg = 0.37$; $SE = 0.04$)” and page 18: “We suggest a negative genetic correlation between LBA and TBV is theoretically more intuitive because a simple measure of TBV will likely capture, among other factors, both early-life neurodevelopment as well as later-life neurodegeneration (i.e., decline which is compatible with a

negative direction of effects” I am not sure whether I agree with this assessment, since the contribution of early life neurodevelopment is probably much larger than the neurodegeneration that follows in terms of the resulting TBV. This expectation seems to be related to the question whether brain atrophy can be expected to be an absolute number (in which case we would not expect a large association between the atrophy measure and the volume, or LBA to be a percentage loss - in which case we would expect larger brains to be associated with larger loss.

We have now amended that sentence as follows, since our initial wording and the reviewer’s point are both speculative and would be important to resolve empirically (line 433):

“We speculate that a negative genetic correlation between LBA and TBV would be theoretically more intuitive where a simple measure of TBV captures, among other factors, both early-life neurodevelopment as well as later-life neurodegeneration (i.e., decline which is compatible with a negative direction of effects). However, this would be less intuitive if neurodevelopment makes a substantially larger contribution to later-life differences in TBV than ageing-related neurodegenerative processes that likely occur over many decades.”

2) Similarly, a thorough discussion about ICV and how it can be measured is lacking. It looks like the authors have taken measures of ICV / TBV at face value. The descriptions of the different cohorts show that in each cohort ICV was measured with different software. Several cohorts use (different versions) of FreeSurfer, and moreover, use eTIV as a measure of ICV. As noted by the creators of the FreeSurfer processing suite, eTIV is not a perfect estimate of ICV, see <https://freesurfer.net/fswiki/eTIV> and its references, especially Klasson, N., Olsson, E., Eckerström, C. et al. Estimated intracranial volume from FreeSurfer is biased by total brain volume. *Eur Radiol Exp* 2, 24 (2018). <https://doi.org/10.1186/s41747-018-0055-4>. The latter manuscript, albeit in a relatively small sample, potentially changes the interpretation of the analyses and results. Given the above, I am not surprised that different cohorts seem to be incomparable.

Thank you for pointing this out.

We have added the *Methods Figures* below (1-2) demonstrating the very high overlap in variance between TBV, ICV and LBA measure in LBC1936 ($N = 581$) when obtained from FreeSurfer v5 vs. FreeSurfer v7 (*SMethods 2.1.3*), illustrating that, at least in the LBC1936, measures have been near identical, had they been obtained with a different FreeSurfer version:

Methods Figure 1. Correlations between TBV, ICV and LBA measures that were obtained with FreeSurfer v5 (FSv5) and FreeSurfer v7 (FSv7)

Methods Figure 2. Associations between equivalent measures extracted from FreeSurfer v5 and FreeSurfer v7

Regarding different FS versions *across* cohorts, in line 233 & 587 we acknowledge software versioning as an inherent challenge that faces large-scale imaging work, particularly in the context of leading international consortia (e.g., ENIGMA), who acknowledge this issue, but proceed under the caveat that they are generally acceptable and within reasonable error margins (e.g., Grasby et al., 2020). We have acknowledged across the manuscript that MRI measures in general have reliability challenges (lines 324, 488, 570). Notably, measurement error in MRI volumetry increases as the ratio between voxel size and the structure being measured increases (smaller structures are more

susceptible to greater measurement error); here we are dealing with estimated volumetry of the largest structures in the brain and thus suffer least from such issues (line 568).

We agree eTIV deserves a thorough discussion which we have added to our Supplementary Methods. This also includes the caveat that eTIV is biased towards TBV as shown by Klasson et al. (2018) (*SMethods*; line 101). Whereas we caution that there are no gold standard measurements (manual ratings and tracings are also subject to forms of error and bias), we report correlations across TBV, ICV, and LBA measures in the LBC1936 ($N = 631$; much larger sample than in Klasson et al.), which were derived based on FreeSurfer (main manuscript), and manually segmented images that are also available in LBC1936.

SMethods 2.1.3: “Details on MRI acquisition and processing software are listed per sample in *Table S1*, indicating that different cohorts were processed with different FreeSurfer software versions. Although it was previously shown that different FreeSurfer software versions can affect study results (e.g., Gronenschild et al., 2012), we had planned this cross-cohort study expecting global measures of TBV and ICV to be within reasonable error margins as is typically assumed by large-scale imaging studies (e.g., Grasby et al., 2020). Within the LBC1936 ($N = 581$), FreeSurfer v5 and FreeSurfer v7 created highly similar interindividual variability for TBV, ICV and LBA measures (*Methods Figures 1-2*). It should be noted that we use the FreeSurfer eTIV measure to approximate ICV. eTIV is inferred through a linear transformation performed to align the image with a given atlas (Buckner et al., 2004). In general, studies consider eTIV a reasonable approximation of ICV. We show here that manual vs. FreeSurfer v5 measures of ICV and TBV showed highly stable individual differences ($r = 0.89$; $r = 0.93$, respectively). Similar to a prior study (Klasson et al., 2018), we find that eTIV is correlated with TBV above and beyond a manually estimated measure of ICV. Correlations between FSv5-based estimates and manually segmented estimates in the LBC1936 ($N = 631$) demonstrate that this bias is present with small effect size ($r_{\text{manually segmented ICV, FS-based LBA residual}} = 0.19$, $p = 1.56 \times 10^{-6}$; *Methods Figure 3*). LBA scores produced by the two estimation techniques yielded substantially correlated variance (LBA_{difference}: $r = 0.68$; LBA_{ratio}: $r = 0.63$, LBA_{residual}: $r = 0.70$; *Methods Figure 4*). Contrary to the FSv5-based CSF estimate, manually-segmented CSF in LBC1936 captured both the ventricle system and the subarachnoid space, which is in line with the substantial correlations between CSF_{manual} and LBA ($r = |0.5-0.96|$).”

Results (line 164): “Analyses in LBC1936 confirm that LBA extracted from different FreeSurfer versions ($N = 581$) or through manual segmentation ($N = 631$) yielded substantially correlated variance, but that using FreeSurfer eTIV to approximate ICV made LBA_{residual} biased towards larger manually-segmented ICV (*SMethods 2.1.3*).”

Discussion (line 502): “Our FreeSurfer-based LBA_{residual} remained modestly biased towards manually-segmented ICV in LBC1936 ($N = 631$), and future studies should test the extent to which this bias pervades other automated methods.”

Discussion (line 586): “Analyses in LBC1936 confirm that LBA extracted from different FreeSurfer versions capture highly similar variance ($N = 581$; *SMethods 2.1.3*). While manual ratings and tracings are also subject to forms of error and bias, the above interpretation must consider that FreeSurfer-based LBA measures show modest bias towards larger manually segmented ICV (shown in LBC1936 $N = 631$; *SMethods 2.1.3*), a challenge that affects most large-scale MRI studies including our genetic findings in UKB.”

Methods Figure 3. Correlations between FreeSurfer v5-based (FS) TBV, ICV and LBA measures and manually segmented TBV, ICV and LBA measures.

Methods Figure 4. Associations between equivalent measures extracted from FreeSurfer v5-based and manually segmented TBV and ICV estimates.

There is a similar issue with the definition of CSF: as far as I am aware, the CSF measure obtained from Freesurfer only captures the ventricle system. Outer CSF - cortical atrophy leading to bigger spaces between the gyri, is not included in this measure; linking LBA to CSF is therefore a very indirect comparison since the latter does not capture cortical atrophy. The authors seem to think it does: Supplementary Methods 2.1.1.1 “considering TBV volume equals ICV volume minus CSF volume.” This strongly depends on the definition of CSF volume.

Thank for pointing out this important point; we had indeed not considered this and removed this inaccurate statement in the Supplementary Methods. The eTIV discussion from above more carefully describes the parts of the brain that are captured by CSF (*SMethods 2.1.3*):

“Contrary to the FSv5-based CSF estimate, manually-segmented CSF in LBC1936 captured both the ventricle system and the subarachnoid space, which is in line with the substantial correlations between CSF_{manual} and LBA ($r = |0.5-0.96|$; *Methods Figure 3*).”

3) Results, LBA phenotype, Section 1: "and below we only report LBA results from the residual method because it consistently outperformed the difference and ratio method. " Based on what is written here, I do not really see a difference between the different LBA measures. One of the strengths of this manuscript is the availability of an expert-defined atrophy measures, but the three LBA measures seem perform similarly. The results in Figure 2 do not really support the conclusion (the intervals of correlations between age-related phenotypes and all three LBA measure largely overlap).

This is an important point that was raised by another reviewer also, and we have amended the statement accordingly (line 192):

“For conciseness, below we only discuss associations for LBA_{residual}, which was similar to those from LBA_{difference} and LBA_{ratio}.”

To be able to make claims about aggregated trends across all traits, we have added a supplementary analysis supporting that all traits (excluding the rating scales) tended to be more associated with LBA_{residual} (line 201) than LBA_{difference} but not LBA_{ratio} (line 205):

“There was a significant aggregated trend to support that LBA_{residual} explained more variance (R^2) across all estimates than LBA_{difference} ($p = 0.0001$; not LBA_{ratio} $p = 0.085$). Though, this was only the case when excluding outlier traits (Cook’s distance $> 4/15$), which were visually rated atrophy (deep and superficial) and brain age; the three cross-sectionally-derived neuroimaging traits included in this analysis (*Fig.S11*; *Fig.S13* for UKB).”

4) Page 13: “extremely healthy nature of UKB participant’s brains compared” this is a strong interpretation - not sure whether it is actually true. Wouldn't this be the result of the different measures used to assess ICV / TBV? (see point 2 above).

We recognise that ‘extremely’ is a value judgement. We have now removed that word from the sentence and replaced it with ‘relatively’.

Another potential explanation is that there seems to be no correction for sex in the phenotypic analyses (specifically relevant for the HCP dataset, page 11 “which seems to have been driven by 303 participants >31 -years with small skulls that were excluded in the HCP data we analysed). Can this negative correlation be caused by a sex effect? Other papers on HCP show that males are overrepresented in the younger ages vs females overrepresented in the older group if I remember correctly.

This is an important point, thank you for raising this. Another reviewer had also asked to include analyses for males and females separately which have now been added to Sections 1-3 in the main manuscript. These analyses show that power is reduced due to the smaller sample size (larger confidence intervals), but that results are largely consistent between analyses of the full sample, males and females only.

Thank you for drawing our attention to this bias in HCP; we were unaware of this. We have investigated this and found that the unexpectedly strong correlation between age and ICV was indeed driven by the females in the sample. We have discussed this and added the plot below to *SMethods*. The plot also confirms that the bias is not present when analysing <31-year-olds only. It is in keeping with this that the 303 participants that were excluded based on our pre-registered age cut-off included two thirds' females, and only one third males.

SMethods 3.1.1: “An anonymous reviewer brought to our attention that the older females in HCP have unusually small ICVs. *Methods Figure 8* below (left) shows that the unexpected association between age and ICV is only significant in females, but not in males. *Methods Figure 8* (right) also shows that this bias is not present when we analyse <31-year-olds only. 203 of the 303 HCP participants that were >31-year and were excluded following the description above were females.”

Methods Figure 8. Sex-split age correlations with ICV. Left: correlation in the full sample; right: correlation in <31-year-olds.

We have also made this explicit in the main manuscript (line 277):

“Note that the raw HCP data showed strong age correlations for ICV (unexpectedly small skulls in older females contradicting our pre-registered expectations; $r = -0.20$, $p = 4.2 \times 10^{-11}$; *SMethods: Deviations from pre-registration*). Following a pre-registered age cut-off, we excluded 303 participants >31-years in the HCP data that we analysed (e.g., in *Fig.3B*).”

5) Comparison of the GWAS results, page 16: "This suggests LBA ratio and LBA difference captured broader and less precise genetic signal compared to LBA residual." Why less precise? Since the measure are slightly different there is no way of knowing which loci should be found here. In addition: the genetic correlations are high and probably not different from 1? I would appreciate a little more information on the genetic loci and their overlap between the LBA measures. What genes

do these loci map to? If the authors ran FUMA, it should be relatively easy to provide a bit more biological context to the GWAS hits.

Upon the reviewer's request, we have altered the wording in the referenced sentence (line 394), and have added six additional supplementary tables (*Tables S10-15*) to list the genomic loci associated with all three LBA phenotypes, and their mapped genes as indicated by FUMA. We have expanded the discussion on the overlap between LBA_{ratio} and LBA_{difference} with LBA_{residual} to include a percentage of independent significant SNPs and mapped genes that overlap between the measures (line 392):

“LBA_{ratio} identified 30 genomic risk loci, 23 of which overlapped with the 28 genomic loci identified by LBA_{residual} in their genomic locations (*Table S12*). LBA_{difference} score identified 37 genomic risk loci, 17 of which overlapped with the 28 LBA_{residual} genomic loci (*Table S14*). This suggests LBA_{ratio} and LBA_{difference} captured broader and less specific genetic signal compared to LBA_{residual}. The top hit in LBA_{ratio} was rs142005327 and the top hit in LBA_{difference} was rs10668066, both located in chromosome 7. The nearest gene for both these SNPs was the *WNT16* gene. All three LBA phenotypes produced similar looking Manhattan plots (*Fig.S21-22*). 71% of the independent significant SNPs and 90% of the mapped genes identified for LBA_{residual} (*Table S11*) were also captured by LBA_{ratio} (*Table S13*). 34% of the independent significant SNPs and 50% of the mapped genes identified for LBA_{residual} were also captured by LBA_{difference} (*Table S15*).”

6) Results, page 17: A genetical correlation of similar size (-0.37 versus 0.35) was described as weak and substantial, respectively. This feels like the authors are a bit biased in their interpretation. Thanks for bringing this inconsistency to our attention. The magnitudes were interpreted relative to their very specific contexts, and we agree this should be reworded to avoid confusion.

First statement (line 413): “LBA_{residual} had a ~~weak~~ moderate negative genetic correlation with our TBV GWAS (r_g residual = -0.37; SE = 0.05)”. This moderate correlation magnitude appeared small compared to the TBV association yielded for LBA_{difference} (r_g difference = 0.37) and LBA_{ratio} (r_g ratio = -0.11).

Second statement (line 422): “Only LBA_{residual} was non-significantly genetically correlated with ICV (r_g = -0.09; SE = 0.05). The other computational methods produced ~~substantial~~ larger and significant genetic correlations with ICV (r_g ratio = 0.35, SE = 0.05; r_g difference = 0.75; SE = 0.06)”. Both the correlation magnitudes for r_g ratio and r_g difference appeared substantial compared to the zero r_g residual.

Other points:

7) The results section is hard to read without referring to the methods (and even though a "methods" section is mentioned several times, I could not find it any of the documents - I received main manuscript, supplementary figures, supplementary methods and supplementary material (including supplementary results and supplementary information for the introduction). Specifically, there is no explanation of the intercepts and slopes in the health measures, apart from a brief mentioning in the STable 2.

We apologise that this was not clear. We have clarified in the manuscript that we refer to the Supplementary Methods ('SMethods') every time we refer to Methods. We have also reworded parts of the results section to give more methodological detail about the growth curve models performed to extract intercepts and slopes (line 174):

“Longitudinally-collected data assessing various cognitive tests, frailty, and body mass index (BMI) were modelled using growth curve models. Intercepts (e.g., *iCog*) were extracted to

capture variance associated with both stable baseline characteristics at age 71, and slopes (e.g., *sCog*) were extracted to capture variance associated with change between the ages 71 to 82 years (*SMethods*).”

Finally, we have included *Table S3* for an overview that summarises all analyses, their aims, analysis code, cohort etc.

8) Introduction: I think it would be good to mention that current genetic studies into brain size or ICV usually assume genetic factors on these phenotypes are the same - sometimes head circumference at birth is combined with adult TBV. I find the footnote on page 4 very relevant - I would not mind this seeing moved to the main text.

This is an interesting point. We have moved the footnote into the introduction, included a reference to Ikram et al. (2012) who state that TBV and ICV reflect *different* genetic influences, which is in line with the approach we are taking too. Finally, we added a reference to Jansen et al. (2020) who meta-analysed SNP effects across TBV in adults and head circumference in children, suggesting that they presume that these phenotypes are good proxies of one another and should therefore yield similar genetic factors.

Line 83: “Supported by evidence that ICV remains broadly stable across the lifespan (Caspi et al., 2020; Holmes et al., 2015; Royle et al., 2013) – which is not true for ageing-related reductions in the volume of the brain – one can employ ICV as an ‘archaeological index’ of an individual’s maximum prior brain size (e.g., Ikram et al., 2012). Using this as a baseline, we may estimate LBA by comparing TBV and ICV in middle-aged and older adults. This concept is distinct from motivations in previous studies that adjust for ICV to control for potentially confounding factors associated with ICV (e.g., Wang et al., 2024), and it applies exclusively to older age cohorts that demonstrate some degree of brain matter shrinkage. Considering TBV with reference to ICV, rather than TBV alone, focuses the analysis on neurodegenerative processes as it amplifies associations with ageing-related traits (e.g., Lyall et al., 2013). Our reasoning does not apply to young cohorts because we cannot reasonably expect brain atrophy in young participants where TBV will be equal to (or very similar to) ICV (e.g., Dhamala et al., 2022). Our approach challenges previous assumptions that TBV in adults and head circumference in children are good proxies of one another (Jansen et al., 2020).”

9) Page 7: " 2) be greater in cross-sectionally scanned older adults (on average), and increase longitudinally with age";

What does increase longitudinally with age mean if not point 3)? Is it different from a linear effect of age on LBA?

We agree the highlighted sentence was not clear enough. This section has been reworded in response to another comment above asking for an introductory paragraph with more methodological detail (line 138 onwards). The former statement (‘be greater in cross-sectionally scanned older adults (on average)’) was commenting on the fact that we hypothesised substantial correlations between LBA and chronological age (Section 2). The latter part of the sentence (‘increase longitudinally with age’) refers to one additional illustration that was moved to the supplement for conciseness showing that LBA gets larger with chronological age in the longitudinally-measured LBC1936 cohort. For clarity, this second statement is now only mentioned once in line 282, and is outlined in the analysis overview *Table S3*.

10) Figure 2: Why was LBA computed at age 82? The authors claim to have no predefined notion of direction or causality - in the comment about APOE - , but I wonder why they try to predict previous

decline (*sCog*, *iCog* - was the former defined at baseline, I would suppose so); using LBA at the oldest age.

This is an important part of our analyses that we should have made clearer. The decision to calculate LBA at age 82 was made to 1) capture the largest amount of atrophy (i.e., more LBA at older ages), and 2) make LBA-trait associations (*Fig.2A&C*) more comparable with trait associations found for longitudinally-observed atrophic changes (*Fig.2B&D*).

Overall, this analytical choice meant that comparisons between LBA and longitudinally-observed atrophic changes (Sections 1&3) overlapped in the time-window that they captured: That is, in LBC1936 both LBA and observed atrophic changes cover brain shrinkage between ages 71 and 82 years. In addition to this 9-year window, LBA also covers an even longer time window which goes back in time until the day each participant first lost any brain matter. This mismatch in timelines likely contributes to dissimilarities between our cross-sectional and longitudinal atrophy estimates (e.g., mentioned in lines 308, 314), but at least they do overlap. Ideally, we would have liked to include an analysis to equate these timelines covered by LBA and longitudinal atrophic changes (and we had pre-registered such an analysis), but this failed due to unexpectedly ‘healthy-looking’ brains in UKB (outlined in line 333 & *SMethods 3.1.2*).

All intercepts and slopes for ageing-related traits in Section 1 (e.g., *sCog* and *iCog*) were extracted from data spanning the same 9-year timeline as the longitudinal atrophic changes, which is likely a contributing reason for the substantial associations between ageing-related traits and longitudinal atrophic changes. We have clarified this in line 174:

“Longitudinally-collected data assessing *cognitive ability*, *frailty*, and *body mass index* (BMI) were modelled using growth curve models. Intercepts (e.g., *iCog*) were extracted to capture variance associated with both stable baseline characteristics at age 71, and slopes (e.g., *sCog*) were extracted to capture variance associated with change between the ages 71 to 82 years (*SMethods*).”

11) Supplement aims 3.2 and 3.3: I am not sure whether creating the plots (and running a regression on) the variance explained is the way to go: by definition there will be more variance explained in the more elaborate model, and the p-value for the extra predictor will indicate whether this is a significant contribution. What is the value of correlation the two or regressing one onto the other? Do you expect exactly the same results for all health related phenotypes?

Yes, indeed we expect R^2 to go up by adding one more predictor to the model, but for this reason we performed a χ^2 test for nested models to understand if adding this second variable significantly improved the model given the loss in degrees of freedom. We also refer the reviewer to our answer to point 1c) above, which outlines the motivations and limitations of the supplementary analyses 3.2 and 3.3. Hopefully, this response addresses part of this point raised here.

We have chosen this approach to get a sense of the variance that a second predictor variable adds to the model in addition to the variance explained by the first predictor variable. We contrast model fit between those nested models using the χ^2 test to ask whether adding this second predictor variable has significantly improved the model given the loss in degrees of freedom, under the null hypothesis that the simple model is better. In analysis 3.3 (LBC1936), the plot indicates R^2 explained by LBA alone (x-axis), and R^2 explained by LBA + longitudinally-estimated changes (y-axis). As pointed out by the reviewer, for most traits (e.g., dementia, *sCog*) we got a significant *p*-value reflecting that longitudinally-estimated changes add variance above and beyond LBA alone. However, this was not the case for other traits like brain age, frailty, and *iCog* where longitudinally-estimated atrophy did *not* add variance above and beyond the variance we explained with LBA (non-significant *p*-value). Hence, this analysis gives us a sense of whether the second predictor variable added to the model significantly. In the UKB, none of the more complex models outperformed the simpler ones.

As in other analyses in this study, we expected an aggregated tendency across all traits that would have indicated how similar (or dissimilar) LBA and longitudinally-observed changes performed in predicting a range of traits. Indeed, if LBA and longitudinally-observed changes had been correlated more highly, we would have expected more non-significant p -values in this analysis, which was, however, not clear in drafting the pre-registration.

12) Discussion: why the use of absolute values? Does not help interpretation.

Thanks for bringing this to our attention. We have revisited the discussion and indicated directions of effects where appropriate (e.g., lines 541, 542). In some instances, we felt that absolute magnitudes were useful when we wanted to indicate the effect size, but the directions were not relevant to the point being made. For example, we have flipped the effect sizes ($*-1$) in $LBA_{residual}$ and LBA_{ratio} to match the directions of effects obtained for $LBA_{difference}$ (i.e., larger values, more brain atrophy). As a consequence, the TBV GWAS by Smith et al. is strongly *negatively* correlated but this is purely because we flipped the effects, and we wanted to avoid confusion when this negative sign was not relevant to the point we are making.

13) Supplementary Table 1: why are all the variances exact multiples of 1000?

Thanks for noticing this. We checked our scripts and can confirm this is due to rounding. We have replaced the multiples of 1000 with numbers rounded to single digits. We have re-ordered the supplementary Tables so that former Table S1 now is Table S6.

14) Supplementary Figure S3: What is STRADL?- it is the Generation Scotland cohort I read from the methods. Please harmonise across the manuscript.

Thanks for bringing this to our attention. STRADL is the acronym for the neuroimaging subset of Generation Scotland. We have now ensured that we refer to ‘Generation Scotland’ across the manuscript (incl. *SFig.3*):

Reviewer #2 (Remarks on code availability):

I have looked at the code and it seems sufficient to assess what was done, I appreciate that the figures from the paper were recreated there. Supplementary data / results do not seem to be included. I have not tried to rerun the analyses using the code.

Thanks for visiting the website, it is good to hear the code is sufficiently annotated. We have now added code for the supplementary analyses also.

Reviewer #3 (Remarks to the Author):

This paper addresses the interesting question of how much information about brain change can be extracted from a one-time point MRI. This was calculated from the relationship between TBV and ICV using different statistical methods and found to correlate modestly with neuroradiological atrophy rating scales ($r = .37-.44$) and longitudinally measured change ($r = .36$). Genetic analyses showed h^2 of 41%, with a strong polygenic signal.

There are many interesting aspects of the study. It addresses a well-known issue, but to my knowledge, it has not been shown with solid empirical data what the size of the correlation between longitudinally and cross-sectionally estimated atrophy actually is. The genetic analyses add to the story in a nice way. Personally, I find this much more convincing (and interesting) than the reported correlation with the neuroradiological rating scale results.

Also, the different relationships of LBA vs. 9-year longitudinal brain change to cognitive decline in the LBC1936 sample are highly interesting. I have never seen anything like that reported previously, directly contrasting current changes vs. estimated accumulated lifetime changes. This is, in my opinion, a very strong aspect of the study!

The different ways of correcting for ICV have been thoroughly investigated before, and the residual approach has been widely regarded as the most optimal one, which aligns with the present results. The LBA measure captures approximately 13% of the variance in actually measured brain change as measured by longitudinal MRIs over 9 years. This is relatively modest but still shows a substantial degree of overlap.

The maximal correlation with longitudinal estimated atrophy will (of course) ultimately depend on the reliability of the measures. Especially for longitudinal estimates, this is a real limitation, particularly for relatively short follow-up intervals as used in, e.g., UKB (see, e.g., <https://www.biorxiv.org/content/10.1101/2024.06.03.592804v1>). I think this is a relevant issue for the current paper; maybe it could shortly be discussed or mentioned?

This is such an important point, thank you for drawing our attention to this pre-print. We clarified our statement about longitudinal findings in UKB in line 323:

‘Given the shorter 4-year time window which likely impacted measurement reliability (Vidal-Piñero et al., 2024), mean atrophic changes were even more limited in UKB (mean $TBV_{time\ 1}$ at age 73 = 1,185 mm^3 , mean $TBV_{time\ 2}$ at age 82 = 1,171 mm^3 ; *SFig.14-15*) than they were in LBC1936.’

We have also emphasised this point and referenced the pre-print in our limitation section.

Line 579: ‘The reliability of longitudinal estimates was likely limited, especially in the UKB where follow-up intervals were short (Brandmaier et al., 2018; Vidal-Piñero et al., 2024).’

The nine-year follow-up interval in the LBC1936 is a real strength in this regard. I think the discussion in general is well-balanced and reflects the nature of the results well, but maybe some more attention could be devoted to the strength of the relationship between the measure of "lifetime atrophy" (LBA) vs. the measure of "current atrophy" (longitudinal change)?

We have added an additional sentence to the first paragraph in the discussion to devote more attention to the strength in association between LBA and longitudinal atrophic changes (line 487):

‘This moderate correlation is noteworthy because it remained robust despite reliability challenges (Vidal-Piñero et al., 2024), and differences in time windows covered by the

measures (i.e., longitudinal atrophic changes cover years, cross-sectional LBA measures cover decades).”

I think the statistics section could include some more information to make the reading easier, as it is presented before the methods. For example, in the analyses of the relationship between LBA and cognitive measures, were any covariates included? Since sex is strongly associated with ICV and TBV, it would be nice to see some results showing that sex did not associate with the different LBA measures (probably does for the difference measure, but maybe not for the residual measure, for example).

We apologise for the lack of clarity. We have restructured the introductory paragraph of the results to include more methodological detail. The edited manuscript now states that no covariates were included to ensure more straightforward comparisons across LBA phenotypes (line 143). We include an additional supplementary table (*Table S3*) to summarise the analyses, interpretations, cohorts, R code etc. Hopefully this will make it more digestible to the reader.

We agree that sex-split analyses are needed to confirm that our results are consistent across males and females. We have repeated analyses presented in Section 1, Section 2 and Section 3 in males and females separately. In Section 1, associations were very similar. One association deviated between males and females in LBC1936: more rapidly increasing frailty levels were only associated with LBA in females, but not males (line 211). Also in the UKB, associations were largely consistent across males and females. The only deviation was that the association between LBA and diabetes was only significant in males but not females (*SFig.7-8*; line 235). In Section 2, results remained stable but yielded fewer significant age correlations when this analysis was repeated in smaller subsamples of males and females separately (*SFig.19*; line 271). In Section 3, we have added *Table S7* to show that associations between LBA and longitudinally-observed atrophic changes were very similar in males and females, with closely overlapping confidence intervals (line 312 & 323).

I was not sure why LBA relations to brain age gaps were reported, as this is another cross-sectional measure, which has been shown to be very weakly related to actual brain change.

We appreciate this comment and agree with this sentiment; it is our feeling that brain age as a measure is still very popularly used in ageing-related studies, and thought it valuable to show how brain age relates to LBA.

A small comment: It is mentioned in the introduction that a problem with longitudinal MRIs is that it is inconvenient and hence limits statistical power, e.g., for genetic studies. I would suggest adding that in a lifespan context, it is currently almost impossible to cover more than a couple of decades using longitudinal MRIs, and a "one-time point" atrophy estimation would then be the only realistically attainable option. The reported change in TBV over 9 years in LBC1936 of approximately 0.75% annually seems perfectly reasonable for this age group, aligning with previous reports of cognitively healthy older adults.

Thank you for this valuable addition. We have added a sentence to reflect this comment in line 64:

‘Given the burden of collecting data over lifespan periods, adopting such a single-occasion approach to modelling neurodegeneration may represent, in some cases, the only feasible option.’

Minor 1: Is the "age-moderated brain shrinkage" mentioned in the abstract the associations with the brain age gap, chronological age, or something else? It would be good to consider another formulation, I believe, as it was not clear to me what this refers to.

Thanks for spotting this inconsistency. The abstract is referring to the LBA associations with chronological age which we have now consistently termed: ‘age-associated brain shrinkage’ (lines 33, 106, 261). In the discussion we have modified the statement to be clearer (line 486): ‘It was also moderated by **chronological** sample age’.

Minor 2: It is stated regarding the genetic analyses that "we performed genome-wide investigations in over 2.5 times more participants." This may strictly speaking be true, but as discussed in the manuscript, other studies have controlled for ICV in the GWAS analyses, which would yield results with some similarities to the residualized approach used here. On the other hand, given the correlation of .36 with actual change, the LBA measure cannot be equated with longitudinal brain change as measured by repeated MRIs. Hence, I am not sure the "2.5-statement" is very relevant.

We understand where the reviewer is coming from and have adjusted the statement accordingly (line 505):

‘Building on this carefully described measure of LBA, we performed **large-scale** genome-wide investigations ($N = 43,110$) **to complement** previous efforts based on meticulously collected longitudinal MRI scans ($N \sim 16,000$; Brouwer et al., 2022).’

Minor 3: Is it accurate to describe $h^2=41\%$ as "strongly heritable"?

Thanks for bringing this to our attention. We had judged this heritability estimate in the context of other brain change heritability estimates which have a tendency to be very small (e.g., Brouwer et al., 2022 which is virtually zero). But we agree that it would be more accurate to refer to this as ‘substantially heritable’ to reflect the numerical estimate without this context, and we have changed the wording accordingly (line 594).

References

- Brandmaier, A. M., Von Oertzen, T., Ghisletta, P., Lindenberger, U., & Hertzog, C. (2018). Precision, reliability, and effect size of slope variance in latent growth curve models: Implications for statistical power analysis. *Frontiers in Psychology, 9*, 294. <https://doi.org/https://doi.org/10.3389/fpsyg.2018.00294>
- Brouwer, R. M., Klein, M., Grasby, K. L., Schnack, H. G., Jahanshad, N., Teeuw, J., Thomopoulos, S. I., Sprooten, E., Franz, C. E., Gogtay, N., Kremen, W. S., Panizzon, M. S., Olde Loohuis, L. M., Whelan, C. D., Aghajani, M., Alloza, C., Alnæs, D., Artiges, E., Ayesa-Arriola, R., . . . the, I. C. (2022). Genetic variants associated with longitudinal changes in brain structure across the lifespan. *Nature Neuroscience, 25*(4), 421-432. <https://doi.org/10.1038/s41593-022-01042-4>
- Buckner, R. L., Head, D., Parker, J., Fotenos, A. F., Marcus, D., Morris, J. C., & Snyder, A. Z. (2004). A unified approach for morphometric and functional data analysis in young, old, and demented adults using automated atlas-based head size normalization: reliability and validation against manual measurement of total intracranial volume. *NeuroImage, 23*(2), 724-738. <https://doi.org/https://doi.org/10.1016/j.neuroimage.2004.06.018>
- Caspi, Y., Brouwer, R. M., Schnack, H. G., van de Nieuwenhuijzen, M. E., Cahn, W., Kahn, R. S., Niessen, W. J., van der Lugt, A., & Pol, H. H. (2020). Changes in the intracranial volume from early adulthood to the sixth decade of life: A longitudinal study. *NeuroImage, 220*, 116842. <https://doi.org/https://doi.org/10.1016/j.neuroimage.2020.116842>
- Dalecki, M., & Willits, F. K. (1991). Examining change using regression analysis: Three approaches compared. *Sociological Spectrum, 11*(2), 127-145.
- Dhamala, E., Ooi, L. Q. R., Chen, J., Kong, R., Anderson, K. M., Chin, R., Yeo, B. T. T., & Holmes, A. J. (2022). Proportional intracranial volume correction differentially biases behavioral predictions across neuroanatomical features, sexes, and development. *NeuroImage, 260*, 119485. <https://doi.org/https://doi.org/10.1016/j.neuroimage.2022.119485>
- Grasby, K. L., Jahanshad, N., Painter, J. N., Colodro-Conde, L., Bralten, J., Hibar, D. P., Lind, P. A., Pizzagalli, F., Ching, C. R. K., McMahon, M. A. B., Shatikhina, N., Zsembik, L. C. P., Thomopoulos, S. I., Zhu, A. H., Strike, L. T., Agartz, I., Alhusaini, S., Almeida, M. A. A., Alnæs, D., . . . group, E. N. G. t. M.-A. C. G. w. (2020). The genetic architecture of the human cerebral cortex. *Science, 367*(6484), eaay6690. <https://doi.org/doi:10.1126/science.aay6690>
- Gronenschild, E. H. B. M., Habets, P., Jacobs, H. I. L., Mengelers, R., Rozendaal, N., van Os, J., & Marcelis, M. (2012). The Effects of FreeSurfer Version, Workstation Type, and Macintosh Operating System Version on Anatomical Volume and Cortical Thickness Measurements. *PLOS ONE, 7*(6), e38234. <https://doi.org/10.1371/journal.pone.0038234>
- Holmes, A. J., Hollinshead, M. O., O'Keefe, T. M., Petrov, V. I., Fariello, G. R., Wald, L. L., Fischl, B., Rosen, B. R., Mair, R. W., Roffman, J. L., Smoller, J. W., & Buckner, R. L. (2015). Brain Genomics Superstruct Project initial data release with structural, functional, and behavioral measures. *Scientific Data, 2*(1), 150031. <https://doi.org/10.1038/sdata.2015.31>
- Ikram, M. A., Fornage, M., Smith, A. V., Seshadri, S., Schmidt, R., Debette, S., Vrooman, H. A., Sigurdsson, S., Ropele, S., Taal, H. R., Mook-Kanamori, D. O., Coker, L. H., Longstreth, W. T., Niessen, W. J., DeStefano, A. L., Beiser, A., Zijdenbos, A. P., Struchalin, M., Jack, C. R., . . . Early Growth Genetics, C. (2012). Common variants at 6q22 and 17q21 are associated with intracranial volume. *Nature Genetics, 44*(5), 539-544. <https://doi.org/10.1038/ng.2245>
- Jansen, P. R., Nagel, M., Watanabe, K., Wei, Y., Savage, J. E., de Leeuw, C. A., van den Heuvel, M. P., van der Sluis, S., & Posthuma, D. (2020). Genome-wide meta-analysis of brain volume identifies genomic loci and genes shared with intelligence. *Nature Communications, 11*(1), 5606. <https://doi.org/10.1038/s41467-020-19378-5>
- Klasson, N., Olsson, E., Eckerström, C., Malmgren, H., & Wallin, A. (2018). Estimated intracranial volume from FreeSurfer is biased by total brain volume. *European Radiology Experimental, 2*(1), 24. <https://doi.org/10.1186/s41747-018-0055-4>

- Privé, F., Luu, K., Blum, M. G. B., McGrath, J. J., & Vilhjálmsson, B. J. (2020). Efficient toolkit implementing best practices for principal component analysis of population genetic data. *Bioinformatics*, 36(16), 4449-4457. <https://doi.org/10.1093/bioinformatics/btaa520>
- Royle, N. A., Booth, T., Valdés Hernández, M. C., Penke, L., Murray, C., Gow, A. J., Maniega, S. M., Starr, J., Bastin, M. E., Deary, I. J., & Wardlaw, J. M. (2013). Estimated maximal and current brain volume predict cognitive ability in old age. *Neurobiol Aging*, 34(12), 2726-2733. <https://doi.org/10.1016/j.neurobiolaging.2013.05.015>
- Rudick, R. A., Fisher, E., Lee, J.-C., Simon, J., Jacobs, L., & Group, t. M. S. C. R. (1999). Use of the brain parenchymal fraction to measure whole brain atrophy in relapsing-remitting MS. *Neurology*, 53(8), 1698-1698. <https://doi.org/10.1212/wnl.53.8.1698>
- Stern, Y., Arenaza-Urquijo, E. M., Bartrés-Faz, D., Belleville, S., Cantilon, M., Chetelat, G., Ewers, M., Franzmeier, N., Kempermann, G., Kremen, W. S., Okonkwo, O., Scarmeas, N., Soldan, A., Udeh-Momoh, C., Valenzuela, M., Vemuri, P., & Vuoksimaa, E. (2020). Whitepaper: Defining and investigating cognitive reserve, brain reserve, and brain maintenance. *Alzheimers Dement*, 16(9), 1305-1311. <https://doi.org/10.1016/j.jalz.2018.07.219>
- van Loenhoud, A. C., Groot, C., Vogel, J. W., van der Flier, W. M., & Ossenkoppele, R. (2018). Is intracranial volume a suitable proxy for brain reserve? *Alzheimer's Research & Therapy*, 10(1), 91. <https://doi.org/10.1186/s13195-018-0408-5>
- Vidal-Piñero, D., Sørensen, Ø., Strømstad, M., Amlien, I. K., Anderson, M., Baaré, W. F. C., Bartrés-Faz, D., Brandmaier, A. M., Bråthen, A. C., Garrido, P., Paolo, G., Grydeland, H., Richard N., H., Kievit, R. A., Kormacher, M., Kühn, S., Lindenberger, U., Mowinckel, A. M., Nyberg, L., . . . Fjell, A. M. (2024). Reliability of structural brain change in cognitively healthy adult samples. *bioRxiv*, 2024.2006.2003.592804. <https://doi.org/10.1101/2024.06.03.592804>
- Wang, J., Hill-Jarrett, T., Buto, P., Pederson, A., Sims, K. D., Zimmerman, S. C., DeVost, M. A., Ferguson, E., Lacar, B., Yang, Y., Choi, M., Caunca, M. R., La Joie, R., Chen, R., Glymour, M. M., & Ackley, S. F. (2024). Comparison of approaches to control for intracranial volume in research on the association of brain volumes with cognitive outcomes. *Human Brain Mapping*, 45(4), e26633. <https://doi.org/https://doi.org/10.1002/hbm.26633>

REVIEWER COMMENTS

Reviewer #1 (Remarks to the Author):

I would still like to see the adjustment method included. Wang concluded this was probably the best method in that it was easiest to implement. Wang also raised significant issues with the residual method. Given arguments in favor of the adjustment method over other methods, including the residual method, it seems like a major omission to exclude it.

We would like to thank the reviewer for raising this point again because we had not expected there to be a notable difference in the results from the adjustment method, and since we were wary of variance inflation. We have experience of large variance inflation when including TBV and ICV in models for cognitive function (e.g., Page et al., 2024). We have now added analyses explicitly comparing results obtained from the residual and adjustment method (*Fig.S40-43*), and indicate variance inflation factor (VIF) values alongside the results. While we can only use an arbitrary VIF cut-off (we chose a cut-off of 5 as it seems the most commonly used; e.g., James et al., 2013) – VIF values indicate that all results reported in *SFig.41* are likely affected by variance inflation, which is also likely the reason why estimates differ so drastically between the residual and adjustment method. A VIF value of 2.25 in *SFig.40* indicates that variance inflation is relatively low, and the results between residual and adjustment method are very similar. We now discuss this across the paper.

Line 181: “Integrating TBV and ICV directly in the model [$lm(outcome \sim TBV + ICV)$] had a variance inflation factor below 5 (VIF = 2.25) and marginally increased effect sizes compared with a two-step residual approach [$LBA_{residual} = resid(lm(TBV+ICV)); lm(outcome \sim LBA_{residual})$] (*SFig.40*).”

Line 271: “Integrating TBV and ICV directly in the model [$lm(age \sim TBV + ICV)$] considerably increased associations with age compared with a two-step residual approach [$LBA_{residual} = resid(lm(TBV+ICV)); lm(age \sim LBA_{residual})$], especially in the UKB and at smaller older-age subsamples. Large variance inflation factor values above 5 suggest that the one-step approach suffers from variance inflation (*SFig.41*).”

Line 393: “Integrating TBV and ICV directly in the GWAS model [$lm(SNP \sim TBV + ICV)$] produced very similar results to the two-step residual approach [$lm(SNP \sim LBA_{residual})$]: mean $\chi^2 = 1.23$; LDSC intercept (SE) = 1.01 (0.008); LDSC heritability (SE) = 0.26 (0.02); independent significant SNPs = 88; loci = 27; *SFig.42*. GWAS derived from both approaches were highly genetically correlated at 0.97 (0.07); the genetic correlation was non-significant between ICV and the adjustment method GWAS ($r_g = 0.046$; $SE = 0.05$).”

Upon publication, we will also make GWAS summary statistics available on Zenodo that were calculated with the adjustment method.

I am not sure I understand why producing two coefficients for the adjustment method is an issue. Regression coefficients are proportional to correlation coefficients and produce the same standard errors. So coefficients can be compared in lieu of correlations without issue.

In the context of our study, we had judged the single $LBA_{residual}$ score desirable for the following reasons: We get an individual-level score which is useful to report descriptive statistics, including sample distributions and other measurement characteristics which was one of the major aims in this paper. This would not have been possible with the adjustment method which, in essence,

is an equation, but does not provide individual-level values. The LBA_{residual} score allowed us to isolate the effect of brain atrophy (defined as the TBV-associated residuals of ICV) at the same time as mitigating the effect of multicollinearity which seemed particularly relevant given TBV and ICV are strongly associated (we had outlined our concern of multicollinearity in the pre-registration, and the large VIF estimates included above also support this). However, we agree with the reviewer that testing effects from the adjustment method, where associations with a third variable were assessed, has been a valuable addition to this paper.

If statistical power is a limitation, a bootstrap could be used in lieu of a conservative z test. I'd maintain that if one wants to discuss differences between methods, one needs to calculate those differences and a corresponding uncertainty measure.

We appreciate this comment, and agree that it would be beneficial to do this had it been the goal to explicitly compare individual associations. However, because it was our goal to model aggregated trends across multiple associations, we believe that the current analyses (as presented) comparing effect sizes on health-related outcomes between LBA_{residual} and both LBA_{ratio} and $LBA_{\text{difference}}$ do not suffer from limitations of statistical power.

Reviewer #2 (Remarks to the Author):

The revision of "Lifetime brain atrophy estimated from a single MRI: measurement characteristics and genome-wide correlates" by Fürtjes and colleagues reads much better than the first version, and the authors really made an effort to answer my questions and add clarification where needed.

I only have a few minor points left:

"Line 441: "It is notable that the residual score was constructed entirely independent of ICV" I am not sure what this means given that LBA_{resid} equals the residual after regressing out ICV?

This statement refers to the fact that LBA_{residual} is, by construction, independent of ICV because it is created as the TBV-associated residuals of ICV [$lm(TBV \sim ICV)$], as outlined in *Box S1*. We have modified the statement to make this clearer:

"It is notable that the residual score is, by construction, entirely independent of ICV (*Box S1*) [...]"

I would suggest the authors take another look at the consistency of the direction of effects. E.g page 18 states ". The same measure of LBA residual was strongly associated with the TBV GWAS by Smith et al. (2021) (rg residual = -0.81; SE = 0.06; negative rg because direction of effects flipped for LBA residual)", while at the beginning they also report a moderate genetic correlation with TBV "LBA residual had a moderate negative genetic correlation with our TBV GWAS (rg residual = -0.37; SE = 0.05)" This one does not seem to be flipped - or it is not stated that this was done, but the direction is the same? The statement "rg across three LBA scores = 0.72-0.96" later on page 18 suggests that the sign was flipped for these analyses.

One solution to all of this would be to mention once that all LBA measures were flipped to indicate "higher values equal more brain atrophy", adjust section 2.2.1 accordingly and adjust signs throughout the paper. This way the absolute values in the discussion are not needed - to be honest, using them still raises red flags for me, as if the signs do not match expectations, which I believe they do - and all other mentions of flipping that make the reader stop and question why could be removed, which would improve readability.

We can confirm that the LBA_{residual} and LBA_{ratio} were flipped *consistently* across the whole manuscript, so that larger values always represent more brain atrophy (e.g., as outlined in lines 229, 279, and 456). For better readability, we have added one more statement to the start of the results section (line 157):

“ LBA_{residual} and LBA_{ratio} were flipped, hence, larger LBA values represent more brain atrophy.”

With regard to the genetic correlations mentioned here, we can confirm that the effects were consistently flipped across all reported genetic correlations. The genetic correlations did indeed indicate that LBA_{residual} is moderately negatively associated with TBV GWAS that were unadjusted for ICV. We speculate in line 417 that TBV likely captures both neurodevelopmental and neurodegenerative variance, and that the overlapping variance here is due to neurodegenerative variance captured by smaller TBV which is associated with more atrophy. Furthermore, we also find that LBA_{residual} is strongly negatively associated with TBV GWAS that were adjusted for ICV, as the approach in Smith et al. mirrors our definition of brain atrophy modelled with the residual score. The only difference is that our score was flipped. Hence, the GWAS by Smith et al. were performed with larger values representing less atrophy – our GWAS was performed on a phenotype where larger values represented more atrophy. We did not flip effects of the Smith et al. GWAS (as we did not calculate them), and instead we remind the reader of the flipped effect sizes in our GWAS (line 404) to explain the negative correlation. We reworded this statement in this second round of revisions to make this clearer (line 405), to complement statements across the paper stating that effect sizes were flipped (lines 233, 287, 418, 470).

Given the reviewer takes issue specifically with absolute values, we have now removed them and indicate the signs of the genetic correlation throughout. The reviewer will see that all directions of correlations indicated in *Fig.5* and the discussion are consistent and sensible.

Reviewer #2 (Remarks on code availability):

I have not reviewed the latest version of the code in detail, it looks complete and well documented.

Reviewer #3 (Remarks to the Author):

The authors have responded to my comments in an excellent way, and the manuscript is in my opinion in ready for publication.

References

- James, G., Witten, D., Hastie, T., & Tibshirani, R. (2013). *An introduction to statistical learning* (Vol. 112). Springer.
- Page, D., Buchanan, C. R., Moodie, J. E., Harris, M. A., Taylor, A., Valdés Hernández, M., Muñoz Maniega, S., Corley, J., Bastin, M. E., Wardlaw, J. M., Russ, T. C., Deary, I. J., & Cox, S. R. (2024). Examining the neurostructural architecture of intelligence: The Lothian Birth Cohort 1936 study. *Cortex*, *178*, 269-286.
<https://doi.org/https://doi.org/10.1016/j.cortex.2024.06.007>